# Phosphate starvation signaling increases mitochondrial membrane potential through respiration-independent mechanisms

Yeyun Ouyang[1], Mi-Young Jeong[1], Corey N Cunningham[1], Jordan A Berg[1], Ashish G Toshniwal[1], Casey E Hughes[1], Kristina Seiler[1], Jonathan G Van Vranken[2], Ahmad A Cluntun[1], Geanette Lam[1], Jacob M Winter[1], Emel Akdogan[3‡], Katja K Dove[1], Sara M Nowinski[1§], Matthew West[4], Greg Odorizzi[4], Steven P Gygi[2], Cory D Dunn[3,5], Dennis R Winge[1,6], Jared Rutter[1,7*§]

[1]Department of Biochemistry, The University of Utah, Salt Lake City, United States; [2]Department of Cell Biology, Harvard University School of Medicine, Boston, United States; [3]Department of Molecular Biology and Genetics, Koç University, İstanbul, Turkey; [4]Department of Molecular, Cellular, and Developmental Biology, University of Colorado, Boulder, Boulder, United States; [5]Institute of Biotechnology, University of Helsinki, Helsinki, Finland; [6]Department of Medicine, The University of Utah, Salt Lake City, United States; [7]Howard Hughes Medical Institute, University of Utah, Salt Lake City, United States

*For correspondence:
rutter@biochem.utah.edu

Present address: ‡Department of Microbiology and Molecular Genetics, University of California, Davis, United States; §Department of Metabolism and Nutritional Programming, Van Andel Institute, Grand Rapids, United States

§Lead Contact

**Abstract** Mitochondrial membrane potential directly powers many critical functions of mitochondria, including ATP production, mitochondrial protein import, and metabolite transport. Its loss is a cardinal feature of aging and mitochondrial diseases, and cells closely monitor membrane potential as an indicator of mitochondrial health. Given its central importance, it is logical that cells would modulate mitochondrial membrane potential in response to demand and environmental cues, but there has been little exploration of this question. We report that loss of the Sit4 protein phosphatase in yeast increases mitochondrial membrane potential, both by inducing the electron transport chain and the phosphate starvation response. Indeed, a similarly elevated mitochondrial membrane potential is also elicited simply by phosphate starvation or by abrogation of the Pho85-dependent phosphate sensing pathway. This enhanced membrane potential is primarily driven by an unexpected activity of the ADP/ATP carrier. We also demonstrate that this connection between phosphate limitation and enhancement of mitochondrial membrane potential is observed in primary and immortalized mammalian cells as well as in *Drosophila*. These data suggest that mitochondrial membrane potential is subject to environmental stimuli and intracellular signaling regulation and raise the possibility for therapeutic enhancement of mitochondrial function even in defective mitochondria.

## Editor's evaluation

Mitochondrial inner membrane potential is a key factor determining several mitochondrial functions, i.e. respiration and protein import, and, thus, affects cellular metabolism. The study identifies a novel mechanism involving phosphate regulation involved in enhancement of inner membrane potential. These fundamental findings are supported by compelling evidence, with rigorous biochemical and state-of-the-art methodology. The results contribute to basic biology knowledge but also open possibilities to modulate mitochondrial potential for therapeutic purposes.

## Introduction

Mitochondria are a central hub for many cellular processes, including ATP production, redox control, biosynthetic programs, and signaling (*Chandel, 2015*; *Pagliarini and Rutter, 2013*; *Spinelli and Haigis, 2018*). Each function—critical for cell homeostasis—relies on the ability of mitochondria to maintain a membrane potential across the inner membrane of this double-membrane organelle. This mitochondrial membrane potential (MMP, $\Delta\Psi$m) directly provides the energy to power ATP synthesis, mitochondrial protein import, and metabolite and ion transport. Either directly or indirectly, it also provides signaling mechanisms to assist in adapting cellular behavior that can be critical to cell health.

Therefore, it is not surprising that impaired MMP is highly correlated with cellular dysfunction in aging (*Hagen et al., 1997*; *Hughes et al., 2020*; *Leprat et al., 1990*; *Mansell et al., 2021*; *Sastre et al., 1996*; *Sugrue and Tatton, 2001*) and a variety of diseases, including primary mitochondrial disease (*Burelle et al., 2015*; *James et al., 1996*) and heart failure (*Cluntun et al., 2021*; *Sharov et al., 2005*). While it is likely that MMP reduction plays a causal role in the pathogenesis of these diseases, the tools to formally test its impact on each disease are limited. In the context of aging, activating an artificial proton pump in *Caenorhabditis elegans* restores the loss of MMP typical of aging and is sufficient to extend lifespan (*Berry et al., 2022*), which suggests causality in this case. In addition, using the same manipulation to ectopically increase MMP improves the survival of *C. elegans* treated with electron transport chain (ETC) inhibitors (*Berry et al., 2020*). These data raise the possibility that low MMP might cause pathology in the context of aging, and perhaps other diseases, and that strategies to restore membrane potential in cells might therefore be therapeutically transformative.

The canonical mechanism to generate MMP is by complexes I, III, and IV of the ETC, which pump protons from the mitochondrial matrix to the intermembrane space (IMS). This intricate process extracts high-energy electrons and passes them through the ETC complexes while using the resultant energy to pump protons from the matrix to IMS. The energy of these protons passing back to the matrix is then used to power ATP synthase, metabolite carriers, and protein translocases. However, several studies have described an alternative mechanism for the generation of MMP, namely ATP synthase running in reverse—hydrolyzing ATP to ADP and using the energy to pump protons to the IMS and augment the MMP (*Junge and Nelson, 2015*; *Okuno et al., 2011*). For example, Vasan et al. reported that the MMP is maintained in complex III-deficient cells through this ATP synthase mechanism (*Vasan et al., 2022*). Despite this long-standing hypothesis, there is conflicting data that argues against this phenomenon (*Vowinckel et al., 2021*). Yet, observations for alternative MMP generation mechanism via ATP hydrolysis illustrate that cells will sacrifice hard-earned ATP to sustain their membrane potential, underlining the essentiality of MMP for cell well-being and insinuating the existence of control mechanisms to maintain MMP (*Ernst et al., 2019*; *Liu et al., 2021*; *Martínez-Reyes et al., 2016*).

MMP is required for viability and proliferation in most eukaryotic cells, but the strength of the MMP is highly variable between cells of different tissue origins and is dynamic across biological conditions (*Huang et al., 2004*; *Mitra et al., 2009*). For example, relative to normal cells, cancer cells tend to have a higher MMP (*Davis et al., 1985*; *Heerdt et al., 2005*; *Summerhayes et al., 1982*) as do cells experiencing amino acid starvation (*Johnson et al., 2014*). Generally, nutrient and other biological stress scenarios also modulate MMP (*Hübscher et al., 2016*, p. 70; *Pan et al., 2011*); in particular, oxidative stress has been shown to decrease MMP (*Korshunov et al., 1997*; *Satoh et al., 1997*). This heterogeneity in MMP amongst cell types and contexts led us to hypothesize that each cell might have an MMP setpoint that is tuned to the energetic and biosynthetic demands of the cell, and is perhaps responsive to nutrients and stressors in the environment. While there are well-appreciated negative consequences when MMP is too low, inappropriately elevated MMP can lead to toxic metabolic byproducts, such as reactive oxygen species. We only have sparse knowledge of whether cells actually have an MMP setpoint. If they do, how is it determined? What are the stimuli that are monitored to determine the setpoint? What are the signaling molecules that communicate this information? How is the machinery of mitochondrial bioenergetics altered to enact the setpoint and maintain this optimal MMP? Answering these questions will provide a much clearer understanding of the connection between cell physiology, mitochondrial bioenergetics, and human disease.

We became interested in MMP and its regulation through our previous studies of the mitochondrial fatty acid synthesis (mtFAS) system. We and others showed that loss of this pathway results in the absence of the lipoic acid cofactor as well as loss of acylated acyl carrier protein (ACP), which

is required for the assembly and activation of many mitochondrial complexes, including each ETC complex (*Angerer et al., 2017*; *Brody et al., 1997*; *Nowinski et al., 2020*; *Van Vranken et al., 2018*). Using a genetic screen in yeast to identify genes required for the transcriptional alterations induced in mtFAS mutants, we found that the deletion of *SIT4* induces high MMP even in the absence of the ETC and ATP synthase. Building on this information, we identified genetic and environmental manipulations that increase MMP via ETC-dependent and -independent mechanisms, including a non-canonical role for the ADP/ATP carrier. These results support the hypothesis that cells leverage available machineries to establish an MMP setpoint that is responsive to internal and external cues. We also identified signaling pathways and molecules that regulate this MMP setpoint. This study describes machinery involved in the modulation of MMP and provides effective tools to better understand the interplay between MMP and cellular health.

## Results

### *SIT4* deletion hyperpolarizes mitochondria

To understand the transcriptional reprogramming that occurs during the loss of mtFAS, and by extension, the interplay between dysfunctional mitochondrial and cell health, we reanalyzed an RNA-sequencing dataset (*Berg et al., 2023*) generated from a yeast mutant lacking mtFAS function (*mct1Δ*) (*Schneider et al., 1997*) before and after transitioning from glucose- to raffinose-containing medium—a manipulation that induces mitochondrial biogenesis. The canonical mitochondrial biogenesis transcriptional response was almost completely absent in the *mct1Δ* cells, as evidenced by the lack of induction of mRNAs encoding subunits of the ETC and ATP synthase (*Figure 1—figure supplement 1A*). Instead, the *mct1Δ* mutant increased mRNA abundance of genes encoding proteins primarily related to mechanisms for acetyl-CoA production—a gene response signature that was not exhibited in wild-type cells (*Figure 1—figure supplement 1B*). These data clearly indicate robust signaling from dysfunctional mitochondria to the nucleus, either to compensate or minimize the damage (*Epstein et al., 2001*; *Garipler et al., 2014*; *Veatch et al., 2009*). To better understand these mitochondria-to-nucleus transcriptional responses, we designed a genetic screen to identify the genes required for the transcriptional aberrations observed in *mct1Δ* cells. We selected one of the most upregulated genes (*CIT2*) to act as a reporter, and a gene with unchanged expression (*BTT1*) to act as a control. We integrated Firefly luciferase at the *CIT2* locus, Renilla luciferase at the *BTT1* locus, and the full-length *CIT2* and *BTT1* genes at the *HO* locus in the *mct1Δ* background (*Figure 1—figure supplement 1C*). We screened through the non-essential gene deletion collection (~5000 genes) using the synthetic genetic array (SGA) methodology (*Giaever et al., 2002*; *Winzeler et al., 1999*). Preliminary hits were reanalyzed using the dual luciferase experiment in a low-throughput manner wherein we could control the optical density of the culture. We identified 73 mutants (displayed in *Figure 1—figure supplement 1D*) with a confirmed reduced ratio of expression from the native *CIT2* and *BTT1* promoters (i.e., reduced Firefly luciferase:Renilla luciferase ratio), suggesting an impaired *mct1Δ* mitochondrial dysfunction transcriptional signature.

We validated all 73 mutants by performing RT-qPCR on four additional genes that were upregulated in *mct1Δ* cells, *DLD3*, *ADH2*, *CAT2*, and *YAT1*, and which we thus use as reporters (*Figure 1—figure supplement 1D*). Some mutants still induced the expression of these genes, and others showed an absence of induction in only a subset of the four target genes (*Figure 1—figure supplement 1D*). Of the mutants that reduced the abundance of all four transcripts, we focused on *SIT4* for two reasons. First, a *sit4Δ* mutant generated in our laboratory, as verification of this screening result, also displayed impaired induction of *CIT2*, *DLD3*, *ADH2*, and *CAT2* in response to *MCT1* deletion (*Figure 1A*). Second, deletion of *SIT4* not only attenuated induction of these four genes, but also induced the expression of several genes encoding ETC subunits, including *QCR2* and *RIP1*, that were repressed in the *mct1Δ* mutant (*Figure 1B*). Notably, deletion of *SIT4* alone—without a concomitant loss of *MCT1*—alters the abundance of these mRNAs tested, indicating that this transcriptional effect of *SIT4* deletion is independent of the underlying mitochondrial defects. These data suggest that *SIT4* contributes to both the positive and negative transcriptional regulation elicited by mitochondrial dysfunction, and therefore we determined to define its role in this context.

*SIT4* is a serine/threonine phosphatase related to human PP6 that plays important roles in cell cycle regulation (*Clotet et al., 1999*; *Fernandez-Sarabia et al., 1992*; *Sutton et al., 1991*), TOR

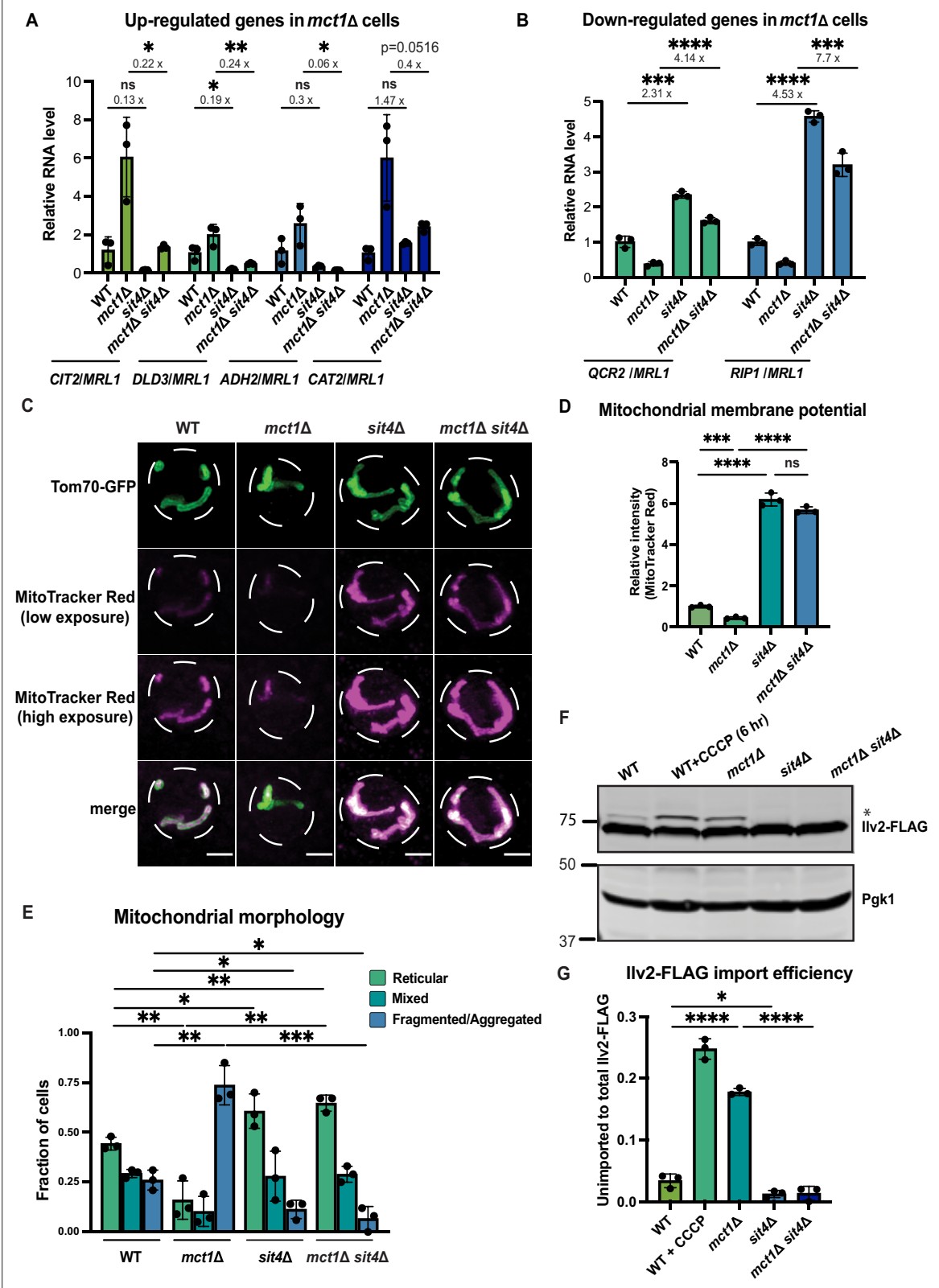

**Figure 1.** *sit4Δ* increases mitochondrial membrane potential in both wild-type and *mct1Δ* cells. (**A, B**) Normalized gene expression of *CIT2*, *DLD3*, *ADH2*, *CAT2*, *QCR2,* and *RIP1* measured 3 hr after switching from media containing 2% glucose as the sole carbon source to media containing 2% raffinose as the sole carbon source. Values were normalized to *MRL1*, a gene that was unchanged by deleting *MCT1* in our RNA-seq dataset. n = 3. Fold changes are displayed. Error bars represent the SD. Statistical significance was determined using an unpaired two-tailed *t*-test. ns = not significant

*Figure 1 continued on next page*

*Figure 1 continued*

p>0.05; *p≤0.05; **p≤0.005; ***p≤0.0005; ****p≤0.0001 (**C**) Representative images of wild-type (WT), *mct1Δ*, *sit4Δ*, and *mct1Δ sit4Δ* strains expressing Tom70-GFP from its endogenous locus stained with MitoTracker Red. Scale bar represents 2 μm. (**D**) Normalized mitochondrial membrane potential of wild-type (WT), *mct1Δ*, *sit4Δ*, and *mct1Δ sit4Δ* strains quantified by flow cytometry measurement of 10,000 cells stained with MitoTracker Red. n = 3. Error bars represent the SD. Statistical significance was determined using an unpaired two-tailed *t*-test. ns = not significant; p>0.05; ***p≤0.0005; ****p≤0.0001. (**E**) Quantification of the fraction of cells in (**C**) showing reticular, mixed, or fragmented/aggregated mitochondrial morphology based on Tom70-GFP signal. n = 3. Error bars represent the SD. Statistical significance was determined using an unpaired two-tailed *t*-test. *p≤0.05; **p≤0.005; ***p≤0.0005. (**F**) Immunoblots of whole-cell lysates extracted from wild-type (WT), *mct1Δ*, *sit4Δ*, and *mct1Δ sit4Δ* strains expressing Ilv2 endogenously tagged with FLAG. As a control, wild-type (WT) cells were treated with 25 μM CCCP for 6 hr. * indicates unimported Ilv2-FLAG. Pgk1 was immunoblotted as a loading control. Original immunoblots are displayed in *Figure 1—source data 1*. (**G**) Normalized quantification of (**F**). Import efficiency is the ratio of unimported (*) to total abundance of Ilv2-FLAG. n = 3. Error bars represent the SD. Statistical significance was determined using an unpaired two-tailed *t*-test. *p≤0.05; ****p≤0.0001. All original immunoblots used for quantification are displayed in *Figure 1—source data 1*.

The online version of this article includes the following source data and figure supplement(s) for figure 1:

**Source data 1.** Source data and uncropped blots used to make *Figure 1*.

**Figure supplement 1.** Genetic screen to identify *SIT4* regulates nuclear responses induced in *mct1Δ* cells.

**Figure supplement 1—source data 1.** Source data and uncropped blots used to make *Figure 1—figure supplement 1*.

signaling (*Rohde et al., 2004*; *Torres et al., 2002*), and tRNA modification (*Abdel-Fattah et al., 2015*). Additionally, a previous study by Garipler et al. showed that deletion of *SIT4* in *rho⁻* cells, where the majority of mitochondrial DNA (mtDNA) is depleted, reverses some of the defects associated with mtDNA damage (*Garipler et al., 2014*). Despite their shared mitochondrial dysfunction, deletion of *MCT1* does not become *rho⁻* as shown by hybrid complementation assays (*Figure 1—figure supplement 1E*).

Based on these gene expression data, as well as previous literature suggesting a role for *SIT4* in regulating OXPHOS (*Garipler et al., 2014*), we sought to understand how *SIT4* affects mitochondrial function and signaling upon loss of mtFAS. First, we measured the MMP in *sit4Δ* cells with or without deletion of *MCT1* using the membrane potential-dependent fluorescent dye, MitoTracker Red, which, when used at the appropriate concentration, accumulates in mitochondria in an MMP-dependent manner, such that the staining positively correlates with MMP. We used both microscopic imaging (*Figure 1C*, *Figure 1—figure supplement 1F*) and flow cytometry (*Figure 1D*) as complementary methods to visualize and quantify MMP. With microscopic imaging, we were able to restrict the quantification of MitoTracker Red signal to that which co-localizes with mitochondria as marked by Tom70-GFP (*Figure 1C*, *Figure 1—figure supplement 1F*). Unexpectedly, we found that cells lacking *SIT4* alone exhibited very high MMP (*Figure 1C and D*, *Figure 1—figure supplement 1F*). The increased MMP observed in *sit4Δ* mutants was also maintained in *mct1Δ sit4Δ* double mutants (*Figure 1C and D*, *Figure 1—figure supplement 1F*). This was surprising given that *mct1Δ* cells lack assembly of the ETC, the major producer of MMP (*Van Vranken et al., 2018*). The *mct1Δ sit4Δ* double mutant also exhibited a modest increase in mitochondrial area (*Figure 1—figure supplement 1G*) and changes in mitochondrial morphology. Therefore, we analyzed the localization pattern of Tom70-GFP and categorized the mitochondrial morphology of each cell as reticular, aggregated/fragmented, or mixed. We found that *mct1Δ* cells exhibited a more fragmented mitochondrial morphology, consistent with previous observations in other respiratory-deficient strains (*Figure 1E*), whereas deletion of *SIT4* resulted in more reticular and less fragmented or aggregated mitochondria, both in the presence or absence of the *mct1Δ* mutation (*Figure 1E*).

The import of many mitochondrial proteins from the cytosol depends upon and thus serves as a proxy for MMP. We used a yeast strain expressing a C-terminally FLAG-tagged Ilv2 at its endogenous locus (*Dasari and Kölling, 2011*). Upon import into mitochondria, the N-terminal mitochondrial targeting sequence (MTS) of Ilv2-FLAG is cleaved, thereby allowing us to distinguish between imported and unimported species and providing a quantitative measurement of Ilv2-FLAG import as assessed by immunoblotting of whole-cell lysates. In wild-type cells, the majority of Ilv2-FLAG is present as a lower molecular weight form with the MTS cleaved (*Figure 1F*); however, a portion of the Ilv2-FLAG protein is visible as a higher molecular weight form, suggesting it has not been cleaved and thus imported into mitochondria. Depletion of the MMP either by treatment for 6 hr with the ionophore CCCP or by deletion of *MCT1* reduced Ilv2-FLAG import and led to accumulation of the uncleaved protein (*Figure 1F and G*). In contrast, deletion of *SIT4* caused a complete loss of

uncleaved Ilv2-FLAG, whether or not *MCT1* was also deleted (*Figure 1F and G*). We interpret these data as suggesting that Ilv2-FLAG is more efficiently imported into mitochondria in the absence of *SIT4*. However, another explanation for the disappearance of the band corresponding to the unimported Ilv2-FLAG is more efficient protein degradation (*Shakya et al., 2021*). Therefore, we sought to more closely examine the relationship between MMP and accumulation of uncleaved Ilv2-FLAG. We used various doses of the protonophore CCCP to dose-dependently reduce the MMP. We found that *sit4Δ* cells treated with 75 μM CCCP have an MMP that is comparable to wild-type cells (*Figure 1— figure supplement 1H*). These cells exhibited a similar level of unimported Ilv2-FLAG as wild-type (*Figure 1—figure supplement 1I and J*). We therefore concluded that it is most likely that the decrease of uncleaved Ilv2-FLAG that we observed in our assays is reflective of increased mitochondrial import and increased MMP. These data demonstrate that loss of *SIT4* results in a mitochondrial phenotype suggestive of an enhanced energetic state: higher MMP, hyper-tubulated morphology, and more effective protein import.

## *SIT4* deletion increases electron transport chain complex abundance

Because the changes in mitochondrial function observed in the *sit4Δ* mutant occur even in the mtFAS mutant background, which on its own creates a profound defect in mitochondrial respiratory complex assembly and energetics (*Van Vranken et al., 2018*), we next asked how the deletion of *SIT4* increases MMP. RNA sequencing revealed that the *sit4Δ* mutant exhibited elevated expression of most of the genes encoding subunits of the ETC and ATP synthase, which could potentially promote ETC complex formation and function (*Figure 2A*, *Supplementary file 2*). To directly assess the abundance of each ETC complex and supercomplexes, we performed blue-native PAGE (BN-PAGE) analysis on mitochondria isolated from wild-type, *mct1Δ*, *sit4Δ*, and *mct1Δ sit4Δ* strains. This enables assessment of the assembly status of respiratory complex II, III, and IV, as well as the ATP synthase. Complex I is not included in our analysis due to its yeast homolog lacking the ability of proton pumping. As previously reported, mitochondria from *mct1Δ* mutant showed a complete loss of all the ETC complexes (*Van Vranken et al., 2018*; *Figure 2B*). Conversely and consistent with the RNA sequencing data, every ETC complex and supercomplex assembly was enriched in the *sit4Δ* mutant (*Figure 2B*). Strikingly, deletion of *SIT4* also completely reversed the absence of and further enriched the abundance of ETC and ATP synthase complexes in *mct1Δ* cells. One potential explanation for this could be the restoration of mtFAS function in the *sit4Δ* mutant; however, *mct1Δ sit4Δ* cells still lacked acylated ACP (*Figure 2—figure supplement 1A*) and lipoic acid as measured by lipoylation of Lat1 and Kgd2 (*Figure 2—figure supplement 1B*), indicating that mtFAS remains inactive upon deletion of *SIT4*. Therefore, we concluded that deletion of *SIT4* enhanced the abundance of ETC complexes, even in the absence of a functional mtFAS pathway.

The enriched ETC complex abundance observed in the *sit4Δ* mutant is similar to what occurs during the glucose de-repression process (*Jin et al., 2007*). We therefore asked whether the increase of MMP and ETC complex assembly could be the result of impaired glucose repression. Normally, yeast cells activate a glucose repression program when grown in media with abundant glucose as a way of optimizing nutrient utilization. During this mode of growth, cells repress mitochondrial biogenesis and rely on glycolysis to provide the ATP required for rapid proliferation. We measured the MMP of wild-type and *sit4Δ* cells grown in glucose medium or in raffinose medium, which causes a loss of glucose repression. As expected from the increased expression of ETC genes, wild-type cells grown in raffinose-containing media had a higher MMP than glucose-grown cells (*Figure 2—figure supplement 1C*). The *sit4Δ* mutant increased MMP to a similar degree in either growth medium (*Figure 2— figure supplement 1C*). This result indicates that the increased MMP caused by *SIT4* deletion cannot be explained by glucose de-repression.

The presence of assembled ETC complexes in *sit4Δ* mutants raises the possibility that they are functional and contributing to MMP via proton pumping. To address this question, we measured the oxygen consumption rate (OCR) in all four yeast strains. As expected, the *mct1Δ* mutant exhibited a profound decrease in OCR (*Figure 2C*). On the other hand, *sit4Δ* cells exhibited an elevated OCR consistent with their increased abundance of ETC complexes. The *mct1Δ sit4Δ* cells showed an OCR that was increased relative to the *mct1Δ* single mutant, but still significantly lower than the wild-type OCR (*Figure 2C*). Thus, we concluded that these ETC complexes are at least partially functional but are not sufficient to rescue the oxygen consumption defects found in the *mct1Δ* mutant. Consistent with

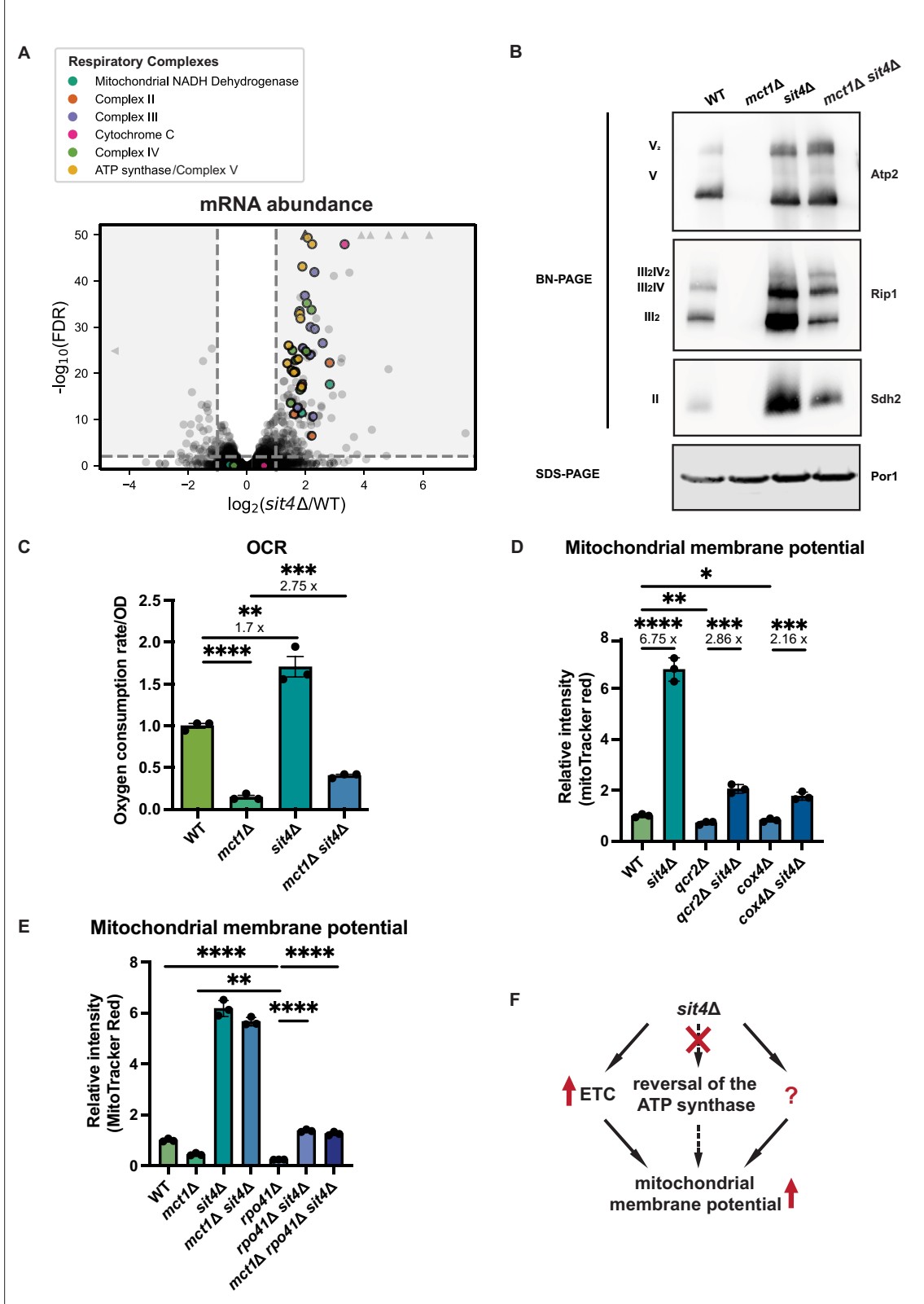

**Figure 2.** *sit4Δ* increases mitochondrial membrane potential through electron transport chain (ETC)-dependent and -independent mechanisms. (**A**) Volcano plot of the transcriptomics data of *sit4Δ* vs. wild-type (WT) cells grown in synthetic media containing 2% glucose. All genes encoding components of the ETC and ATP synthase that were detected by RNA sequencing are highlighted and color-coded if they passed the detection and analysis criteria. Triangle indicates that the -log10 (FDR) exceeds 50. (**B**) Immunoblots of crude mitochondria extracted from wild-type (WT), *mct1Δ*, *sit4Δ*,

*Figure 2 continued on next page*

*Figure 2 continued*

and *mct1Δ sit4Δ* strains and separated on both BN-PAGE or SDS-PAGE. Membranes were blotted with indicated antibodies. Por1 was immunoblotted as a loading control. Original immunoblots are displayed in *Figure 2—source data 1*. (C) Normalized oxygen consumption rate (OCR) over optical density (OD) of the indicated strains grown in synthetic media containing 2% raffinose. n = 3. Error bars represent the SD. Statistical significance was determined using an unpaired two-tailed *t*-test. **p≤0.005; ***p≤0.0005; ****p≤0.0001. (D, E) Normalized mitochondrial membrane potential of wild-type (WT), *sit4Δ*, *qcr2Δ*, *qcr2Δ sit4Δ*, *cox4Δ*, *cox4Δ sit4Δ*, *mct1Δ*, *rpo41Δ*, *rpo41Δ sit4Δ*, and *mct1Δ rpo41Δ sit4Δ* strains quantified by flow cytometry measurement of 10,000 cells stained with MitoTracker Red. n = 3. Error bars represent the SD. Statistical significance was determined using an unpaired two-tailed *t*-test. *p≤0.05; **p≤0.005; ***p≤0.0005; ****p≤0.0001. (F) Schematic of mechanisms through which *sit4Δ* increases mitochondrial membrane potential. The mechanism (reversal of ATP synthase) that is theoretically possible but not utilized in *sit4Δ* cells is marked in dashed line.

The online version of this article includes the following source data and figure supplement(s) for figure 2:

**Source data 1.** Source data and uncropped blots used to make *Figure 2*.

**Figure supplement 1.** *SIT4* deletion does not restore ACP acylation or respiratory growth in *mct1Δ* cells.

**Figure supplement 1—source data 1.** Source data and uncropped blots used to make *Figure 2—figure supplement 1*.

previous reports (*Arndt et al., 1989*; *Dimmer et al., 2002*; *Jablonka et al., 2006*), *sit4Δ* cells failed to grow on media containing a non-fermentable carbon source such as glycerol that requires mitochondrial respiration (*Figure 2—figure supplement 1D*). This defect was rescued by re-expression of *SIT4* on a plasmid, confirming that *sit4Δ* cells do not have an irreversible loss of mtDNA as would be observed in a *rho⁰* stain. As expected, the *mct1Δ sit4Δ* double mutant also failed to grow under respiratory conditions (*Figure 2—figure supplement 1D*). In conclusion, deletion of *SIT4* promotes the assembly of partially functional ETC complexes, but this is insufficient to rescue the respiratory defects of the *mct1Δ* mutant. Moreover, the observed modest increase in oxygen consumption is insufficient to explain the profound increase in MMP observed in the *mct1Δ sit4Δ* double mutant relative to the *mct1Δ* single mutant.

The yeast Hap complex has been shown to transcriptionally induce the expression of OXPHOS components (*Bonander et al., 2008*; *Mao and Chen, 2019*). We assessed whether increased expression of OXPHOS components by the Hap complex might be sufficient to increase MMP similar to what we observed in *sit4Δ* cells. Overexpression of *HAP4*, the catalytic component of the HAP complex, was sufficient to increase the abundance of each OXPHOS complex as assessed by BN-PAGE (*Figure 2—figure supplement 1E*). However, *HAP4* overexpression failed to rescue the loss of any of the OXPHOS complexes observed in the *mct1Δ* background (*Figure 2—figure supplement 1E*). It also had no effect on MMP in either wild-type or *mct1Δ* cells (*Figure 2—figure supplement 1F*). Loss of *HAP4* suppressed, but did not eliminate, the increased MMP observed in *sit4Δ* cells (*Figure 2—figure supplement 1G*). Therefore, we concluded that *sit4Δ* cells partially require the Hap complex to generate higher MMP, but overexpressing *HAP4* is insufficient to increase MMP.

As a result of the OXPHOS complex abundance and activity data, we asked whether the enhanced MMP in *sit4Δ* mutants was dependent upon ETC proton pumping. We deleted subunits that are necessary for complex III (*QCR2*) and IV (*COX4*) assembly and function in both wild-type and *sit4Δ* mutant strains. Whether or not complex III or IV was inactivated, deletion of *SIT4* was sufficient to increase MMP, albeit not to the same extent as in wild-type cells (*Figure 2D*). As mentioned previously, in the absence of a functional ETC, the mitochondrial ATP synthase has been shown to reverse direction and use the energy of ATP hydrolysis to pump protons across the mitochondrial inner membrane and generate a membrane potential. We treated wild-type cells and the *sit4Δ* mutant with oligomycin, an inhibitor of the $F_o$ portion of the ATP synthase, to examine the directionality of the ATP synthase activity. Inhibiting the ATP synthase led to an increase of MMP in both wild-type and *sit4Δ* cells (*Figure 2—figure supplement 1H*), suggesting that ATP synthase operates in the direction of MMP consumption and ATP generation in *sit4Δ* cells.

To orthogonally test whether the *sit4Δ* mutant can use mechanisms independent of ETC and ATP synthase, we obtained *rho⁰* cells that completely lack mtDNA and therefore have no functional ETC or ATP synthase. In spite of repeated efforts and success generating *rho⁰* cells in wild-type or other mutant backgrounds, we were unable to generate *rho⁰* cells in a *sit4Δ* strain. Therefore, we genetically eliminated the mitochondrial RNA polymerase, *RPO41*, which is required to transcribe all mtDNA-encoded transcripts (*Greenleaf et al., 1986*; *Wang and Shadel, 1999*). As a result, the *rpo41Δ* mutant lacks the proton-pumping ETC complexes III and IV as well as the protein-transducing $F_o$ component of the ATP synthase (*Figure 2—figure supplement 1I*). As expected, the MMP of the *rpo41Δ* mutant

is very low, even significantly lower than that of *mct1Δ* cells (*Figure 2E*). However, deletion of *SIT4* in the *rpo41Δ* mutant background restored the MMP to a level similar to wild-type cells (*Figure 2E*). Intriguingly, complex II, which does not contain any mtDNA-encoded subunits, accumulated more assembled complex upon *SIT4* deletion (*Figure 2—figure supplement 1I*). These results demonstrated that although ETC activity is required for the majority of the enhanced MMP observed in *sit4Δ* cells, *sit4Δ* mutants clearly leverage additional ETC- and ATP synthase-independent mechanisms to increase mitochondrial membrane potential (*Figure 2F*).

## The *sit4Δ* mutant exhibits a phosphate starvation response

To discover the mechanisms through which the *sit4Δ* mutants generate MMP independent of the ETC or ATP synthase, we performed phosphoproteomics, as Sit4 is a protein phosphatase. Among the most enriched phosphoproteins, Pho84, Vtc3, and Spl2 are all involved in the regulation of intracellular phosphate levels (*Figure 3A*, *Supplementary file 3*). Pho84 is a high-affinity phosphate transporter on the plasma membrane, Vtc3 is involved in polyphosphate synthesis, and Spl2 mediates the downregulation of the low-affinity phosphate transporter during phosphate depletion. In addition, we cross-referenced our phosphoproteomics and RNA sequencing data (*Figure 2A*, *Supplementary file 2*) and confirmed that many of the most significant transcriptional increases in the *sit4Δ* mutant were transcriptional targets of the PHO regulon (*Figure 3B*), which stimulates phosphate acquisition triggered by either phosphate depletion or by dysregulation of the phosphate signaling pathway (*Oshima, 1997*; *Paolo et al., 1997*).

## Phosphate depletion increases mitochondrial membrane potential in wild-type and *rho⁰* cells

Given the unexpected activation of a phosphate starvation response upon deletion of *SIT4*, we tested whether modulation of environmental phosphate abundance could directly increase MMP. All organisms acquire phosphate from the environment to build nucleic acids, phospholipids, and phosphorylated sugars and proteins. The synthetic media commonly used for growing yeast contains 7.5 mM inorganic phosphate. We depleted phosphate by growing yeast cells in media with close-to-normal phosphate levels (10 mM) or a series of lower phosphate concentrations (100, 50, 10, or 1 µM) and then measured the MMP using flow cytometry. It is important to note that all media were buffered at the same pH (pH 4.1). Over an 8 hr time course, we observed an increase in the MMP of wild-type cells grown in low phosphate concentrations (50, 10, or 1 µM), but not in cells grown in either 100 µM or 10 mM phosphate (*Figure 3C*). Having found that *sit4Δ* cells exhibited increased MMP even in the absence of the ETC and the F$_o$ subunit of ATP synthase (*Figure 2E*), we performed a similar phosphate depletion experiment in *rho⁰* cells, which lack all components of these two proton-pumping systems. Indeed, *rho⁰* cells exhibited elevated MMP in response to phosphate depletion (1 µM Pi), although the response was delayed compared to wild-type cells (*Figure 3D*). We also confirmed this observation using fluorescence microscopy to capture the MMP and mitochondrial morphology after depleting phosphate in the media for 4 hr in wild-type cells and overnight in *rho⁰* cells (*Figure 3—figure supplement 1A*). The quantification of the MitoTracker Red signal co-localized within Tom70-GFP agreed with the measurement of MMP using flow cytometry (*Figure 3—figure supplement 1B*). At the same time, no significant change in mitochondrial area was observed by depleting phosphate from the media (*Figure 3—figure supplement 1C*). Finally, the quantification of mitochondrial morphology suggested that phosphate deprivation does not induce massive changes in mitochondrial morphology in wild-type cells (*Figure 3—figure supplement 1D*). However, the fragmented and aggregated mitochondria in *rho⁰* cells was partially rescued by phosphate depletion (*Figure 3—figure supplement 1D*). Consistent with the MMP increase, mitochondrial protein import, as measured by Ilv2-FLAG cleavage, was enhanced by phosphate depletion in wild-type cells (*Figure 3E and F*). *rho⁰* cells have a significant import deficiency, but this defect was mostly rescued by phosphate depletion as well (*Figure 3E and F*).

Next, we examined respiratory complex assembly to determine additional ETC and ATP synthase-related factors contributing to MMP. Using BN-PAGE, we found that in wild-type cells, phosphate depletion led to a modest enrichment in ETC complex abundance compared to cells grown in normal phosphate concentrations (*Figure 4A*). Inversely, as expected, depleting phosphate in *rho⁰* cells failed to rescue the absence of the complexes. Similar to the complex II enrichment in *rpo41Δ sit4Δ* cells

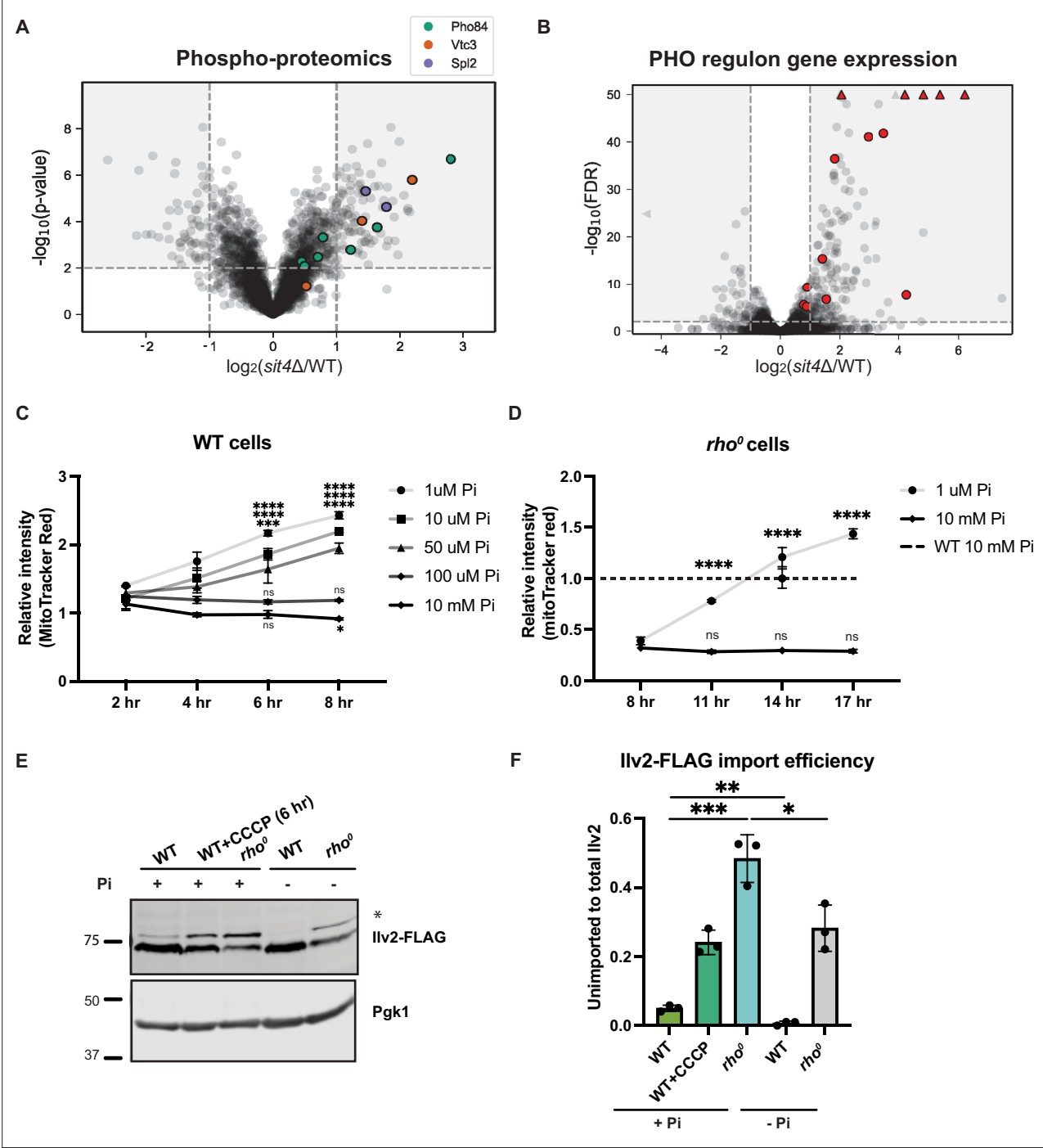

**Figure 3.** Phosphate starvation increases mitochondrial membrane potential through electron transport chain (ETC)-dependent and independent mechanisms. (**A**) Volcano plot of phosphoproteomics data of *sit4Δ* vs. wild-type (WT) cells grown in synthetic media containing 2% glucose. Unique phosphorylation sites of Pho84 (green), Vtc3 (red), and Spl2 (purple) are highlighted. (**B**) Volcano plot of transcriptomics data of *sit4Δ* vs. wild-type (WT) cells grown in synthetic media containing 2% glucose. All PHO regulon targets that are detected by RNA sequencing are highlighted in red. Triangle indicates that the -log10 (FDR) exceeds 50. (**C, D**) Normalized time course of mitochondrial membrane potential in wild-type (WT) and *rho⁰* strains measured by flow cytometry. The dashed line represents the membrane potential of wild-type (WT) cells grown in media containing 10 mM phosphate. n = 3. Fold changes are displayed. Error bars represent the SD. Statistical significance was determined using two-way ANOVA with Tukey's multiple comparisons. ns = not significant; p>0.05; ***p≤0.0005; ****p≤0.0001. (**E**) Immunoblots of whole-cell lysates extracted from wild-type (WT) or *rho⁰* cells expressing Ilv2 endogenously tagged with FLAG. Wild-type and *rho⁰* cells were grown in media containing either 10 mM of phosphate (+Pi) or 1 μM of phosphate (-Pi) for 4 hr or overnight, respectively. As a control, wild-type (WT) cells were treated with 25 μM CCCP for 6 hr. * indicates unimported Ilv2-FLAG. Pgk1 was immunoblotted as a loading control. Original immunoblots are displayed in *Figure 3—source data 1*. (**F**) Normalized quantification

*Figure 3 continued on next page*

*Figure 3 continued*

of (**E**). Import efficiency is the ratio of unimported (*) to total abundance of Ilv2-FLAG. n = 3. Error bars represent the SD. Statistical significance was determined using an unpaired two-tailed *t*-test. *p≤0.05; **p≤0.005; ***p≤0.0005. All original immunoblots used for quantification are displayed in *Figure 3—source data 1*.

The online version of this article includes the following source data and figure supplement(s) for figure 3:

**Source data 1.** Source data and uncropped blots used to make *Figure 3*.

**Figure supplement 1.** Phosphate depletion increases mitochondrial membrane potential in wild-type and *rho⁰* cells.

(*Figure 2—figure supplement 1I*), phosphate depletion also led to an accumulation in assembled complex II in *rho⁰* cells (*Figure 4A*).

We then treated cells with a series of mitochondrial inhibitors to parse out contributions of the ETC, ATP synthase, and other mechanisms to the MMP. Because electron transfer through complexes III and IV is tightly coupled to one another and with proton pumping, the complex III inhibitor antimycin A (AA) is sufficient to block the activities of both complexes. Most of the membrane potential was lost in cells treated with antimycin A in either normal or low phosphate-containing media (*Figure 4B*); however, even in the presence of antimycin A, low phosphate still triggered a similarly fold elevation in membrane potential. Together with the BN-PAGE results, we concluded that ETC complexes are slightly enriched with phosphate depletion, but this is not required for the increase in MMP.

Bongkrekic acid is an inhibitor of the ADP/ATP carrier (AAC) (*Lauquin and Vignais, 1976*), which resides on the mitochondrial inner membrane and normally imports ADP and exports ATP to sustain mitochondrial ATP synthesis and cytosolic ATP consumption. Treatment of wild-type cells with bongkrekic acid significantly dampened the phosphate depletion-mediated increase in MMP (*Figure 4B*). Importantly, combination treatment with both antimycin A and bongkrekic acid completely blocked the induction of MMP in response to low phosphate (*Figure 4B*). As a genetic alternative to antimycin A inhibition of the ETC, we grew *rho⁰* cells, which have no complex III and IV nor a complete ATP synthase, in low and high phosphate (*Figure 4C*). As shown before, phosphate depletion triggers an enhanced MMP in *rho⁰* cells, but this is completely eliminated by bongkrekic acid in a dose-dependent manner (*Figure 4C*).

These experiments suggest a mechanism whereby the depletion of phosphate increases MMP in an ETC- and ATP synthase-independent manner. When the ADP/ATP carrier imports $ATP^{4-}$ and exports $ADP^{3-}$, a net export of a positive charge occurs out of the matrix to the IMS (*Figure 4D*). This activity must be coupled to ATP hydrolysis within the mitochondrial matrix. It would also be coupled to the export of phosphate from the matrix, which is co-transported with a proton through the phosphate carrier. Our data suggest that when cells lack the proton-pumping ability of the ETC—either by chemical treatment (i.e., antimycin A) or genetic inhibition (mtDNA loss in *rho⁰* cells)—and particularly during phosphate depletion, they instead rely on the ADP/ATP carrier to increase MMP to sustain critical mitochondrial functions. We refer to this alternative mechanism to generate MMP as the mitochondrial ATP hydrolysis pathway.

## Phosphate starvation signaling induces mitochondrial membrane potential

The phosphate signaling system in yeast (*Kaffman et al., 1994*; *Mouillon and Persson, 2006*) employs the cyclin and cyclin-dependent kinase, Pho80 and Pho85, respectively. Under normal phosphate conditions, Pho85 is active but it is inhibited during phosphate depletion. When active, Pho85 phosphorylates and inactivates Pho4, a transcriptional factor that stimulates PHO regulon genes to promote phosphate acquisition, maintenance, and mobilization (*Figure 5—figure supplement 1A*).

Deletion of *PHO85* in yeast cells results in constitutive activation of the PHO regulon even in a high phosphate environment. As a result, *pho85Δ* cells accumulate twice as much phosphate as wild-type cells (*Gupta et al., 2019*; *Liu et al., 2017*). To elucidate how environmental phosphate starvation induces high MMP, we measured the MMP in *pho85Δ* mutants and found that, similar to *sit4Δ* cells or cells grown in phosphate-depleted media, MMP was significantly increased in *pho85Δ* cells (*Figure 5A*, *Figure 5—figure supplement 1B and C*). In the context of *MCT1* deletion, the *pho85Δ* also increased MMP, restoring the *mct1Δ* to near wild-type membrane potential (*Figure 5A*, *Figure 5—figure supplement 1B and C*). Deletion of *PHO85* had no effect on mitochondrial area

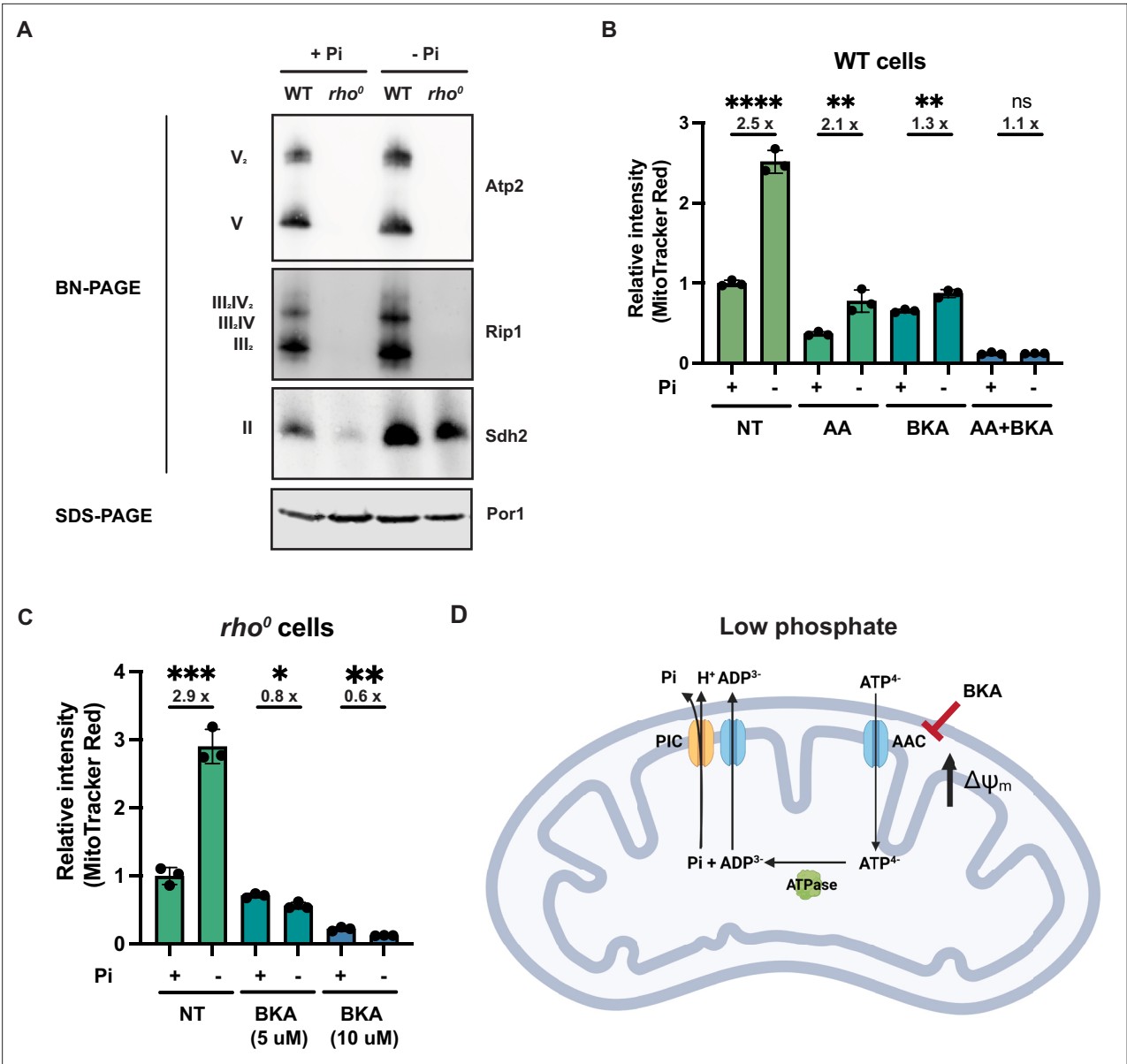

**Figure 4.** Phosphate depletion promotes mitochondrial membrane potential via ADP/ATP carrier in cells without electron transport chain (ETC) and ATP synthase. (**A**) Wild-type (WT) and *rho⁰* cells were grown in 10 mM (+Pi) or 1 μM (-Pi) phosphate-containing media for 4 hr or overnight, respectively. Crude mitochondria were extracted and separated by BN-PAGE or SDS-PAGE. Membranes were blotted with the indicated antibodies. Por1 was immunoblotted as a loading control. Original immunoblots are displayed in *Figure 4—source data 1*. (**B**) Wild-type (WT) cells were grown in 10 mM (+Pi) or 1 μM (-Pi) phosphate-containing media with or without drug treatment for 4 hr. Mitochondrial membrane potential was measured and quantified by flow cytometry. Fold changes are displayed. Error bars represent the SD. Statistical significance was determined using an unpaired two-tailed *t*-test. ns = not significant; $p > 0.05$; **$p \leq 0.005$; ****$p \leq 0.0001$. (**C**) *rho⁰* cells were grown in 10 mM (+Pi) and 1 μM (-Pi) phosphate-containing media overnight and treated with or without bongkrekic acid (BKA) for 4 hr. Mitochondrial membrane potential was quantified by flow cytometry measurements of MitoTracker Red. Fold changes are displayed. Error bars represent the SD. Statistical significance was determined using an unpaired two-tailed *t*-test. *$p \leq 0.05$; **$p \leq 0.005$; ***$p \leq 0.0005$. (**D**) Schematic of the mechanisms for increased mitochondrial membrane potential induced by phosphate depletion.

The online version of this article includes the following source data for figure 4:

**Source data 1.** Source data and uncropped blots used to make *Figure 4*.

(*Figure 5—figure supplement 1D*). In spite of having an elevated MMP, the *pho85Δ* mutant shared a similar distribution of reticular, fragment/aggregated, or mixed mitochondrial morphology with wild-type cells (*Figure 5B*). However, *PHO85* deletion normalized the fragmented and aggregated mitochondrial phenotype of *mct1Δ* cells (*Figure 5B*). We also confirmed the MMP observation by

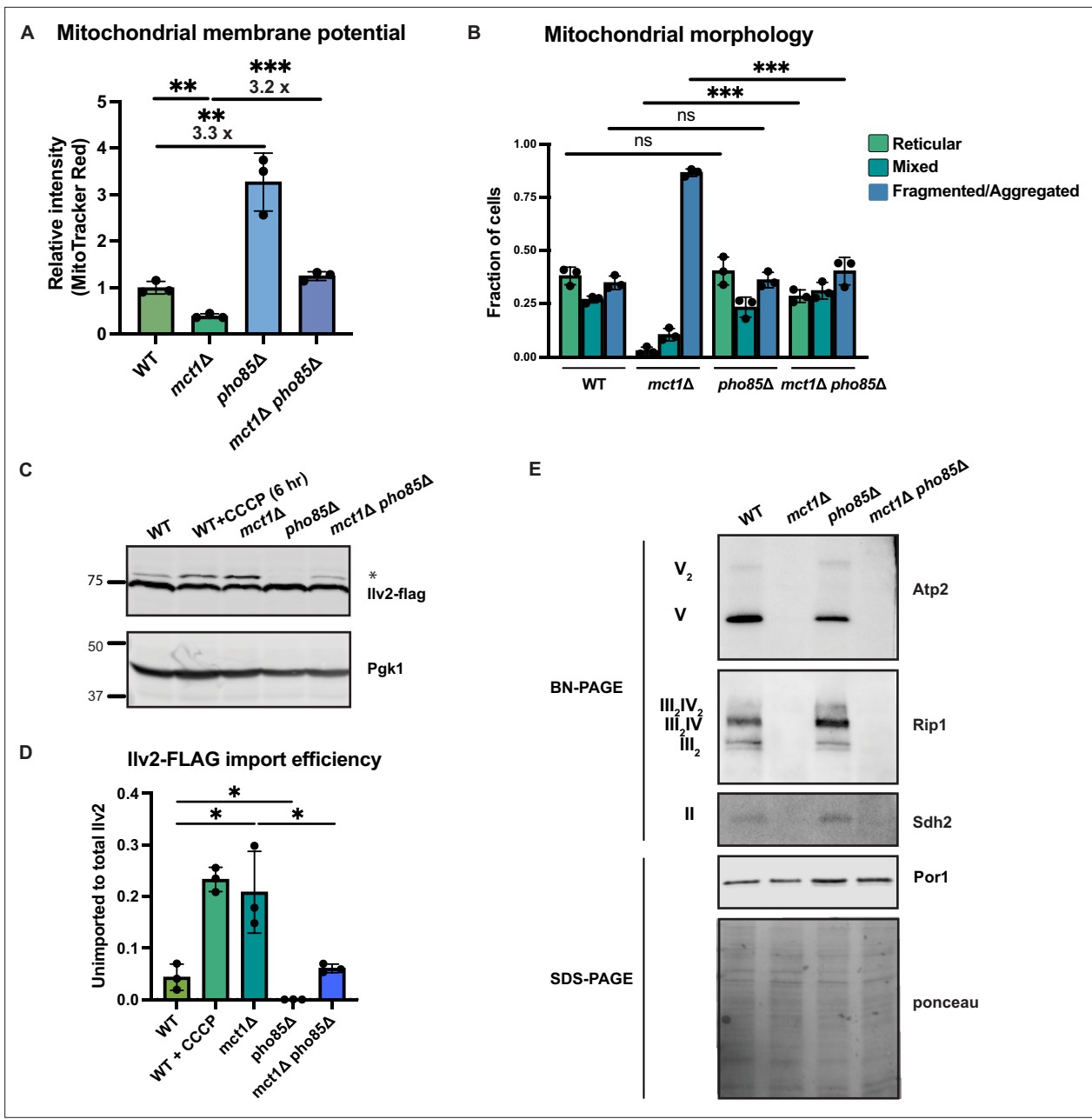

**Figure 5.** Activation of phosphate signaling increases mitochondrial membrane potential. (**A**) Normalized mitochondrial membrane potential of wild-type (WT), *mct1Δ*, *pho85Δ*, and *mct1Δ pho85Δ* strains quantified by flow cytometry measurement of 10,000 cells stained with MitoTracker Red. n = 3. Fold changes are displayed. Error bars represent the SD. Statistical significance was determined using an unpaired two-tailed *t*-test. **p≤0.005; ***p≤0.0005. (**B**) Quantification of the fraction of cells in *Figure 4B* showing reticular, mixed, or fragmented/aggregated mitochondrial morphology based on Tom70-GFP signal. n = 3. Error bars represent the SD. Statistical significance was determined using an unpaired two-tailed *t*-test. ns = not significant; p>0.05; ***p≤0.0005. (**C**) Immunoblots of whole-cell lysates extracted from wild-type (WT), *mct1Δ*, *pho85Δ*, and *mct1Δ pho85Δ* cells expressing Ilv2 endogenously tagged with FLAG. As a control, wild-type (WT) cells were treated with 25 µM CCCP for 6 hr. * indicates unimported Ilv2-FLAG. Pgk1 was immunoblotted as a loading control. Original immunoblots are displayed in *Figure 5—source data 1*. (**D**) Normalized quantification of (**C**). Import efficiency is the ratio of unimported (*) to total abundance of Ilv2-FLAG. n = 3. Error bars represent the SD. Statistical significance was determined using an unpaired two-tailed *t*-test. *p≤0.05. All original immunoblots used for quantification are displayed in *Figure 5—source data 1*. (**E**) Immunoblots of crude mitochondria extracted from wild-type (WT), *mct1Δ*, *pho85Δ*, and *mct1Δ pho85Δ* cells and separated by BN-PAGE or SDS-PAGE. Membranes were blotted with indicated antibodies. The membrane was stained with Ponceau S and blotted with Por1 antibody as loading controls. Original immunoblots are displayed in *Figure 5—source data 2*.

The online version of this article includes the following source data and figure supplement(s) for figure 5:

**Source data 1.** Source data and uncropped blots used to make *Figure 5*.

*Figure 5 continued on next page*

*Figure 5 continued*

**Source data 2.** Source data and uncropped blots used to make *Figure 5*.

**Figure supplement 1.** Phosphate starvation signaling increases mitochondrial membrane potential.

demonstrating that the *pho85Δ* mutant had improved import of Ilv2-FLAG than wild-type cells, and *PHO85* deletion restored the import activity of the *mct1Δ* mutant to around wild-type levels (*Figure 5C and D*). These phenotypes of the *pho85Δ* mutant strain suggest that the mechanism underlying the MMP increase in response to phosphate starvation is unlikely to be a direct result of intracellular phosphate insufficiency. Rather, these data support the hypothesis that the perception of phosphate starvation, and activation of the phosphate signaling response, is the key driver of MMP enhancement.

Next, we asked whether deletion of *PHO85* could rescue the ETC complex assembly defect in the *mct1Δ* mutant, as observed in the *sit4Δ* mutant, which could be an explanation for the enhanced MMP. Consistent with phosphate-depleted wild-type cells as shown in *Figure 4A*, there was a modest enrichment of ETC complexes in the *pho85Δ* mutant (*Figure 5E*). In the context of the *mct1Δ* mutant strain, however, deletion of *PHO85* had no effect on the complete loss of ETC complexes. We reasoned that this lack of rescue was different from *SIT4* deletion because *pho85Δ* cells exhibit no increase in the abundance of mRNAs encoding ETC and ATP synthase subunits (*Figure 5—figure supplement 1E*, compared to *Figure 2A* for *sit4Δ*, *Supplementary file 2*). We therefore concluded that activating phosphate starvation signaling is sufficient to establish an elevated MMP, in a manner that is mostly independent of the ETC. In particular, *mct1Δ pho85Δ* cells exhibit a much higher MMP compared to the *mct1Δ* mutant, and this appears to largely be mediated by ETC- and ATP synthase-independent mechanisms.

We demonstrated that phosphate depletion utilizes mainly the mitochondrial ATP hydrolysis pathway, via the exchange of differentially charged nucleotides, to generate MMP. Similar models have been proposed previously, and it was suggested that the $F_1$ subunit of the ATP synthase (Atp1 and Atp2) catalyzes the ATP hydrolysis (*ATP synthase of yeast mitochondria—GIRAUD - 1994, 2023*; *Lefebvre-Legendre et al., 2003*). To test this model, we attempted to generate combination mutants of *atp1Δ* or *atp2Δ* along with *mct1Δ* or *pho85Δ*. It was not surprising that *atp1Δ mct1Δ* and *atp2Δ mct1Δ* cells were not viable (*Figure 5—figure supplement 1F and G*) based on the synthetic lethality observed upon *ATP1* deletion in *rho⁰* cells. However, deleting *PHO85* partially rescued the lethality of the double mutant (*Figure 5—figure supplement 1F and G*). These data suggest that additional ATPase(s) beyond the $F_1$ subunit of ATP synthase couple with ADP/ATP carrier activity to generate MMP in the phosphate-depleted condition.

## Depleting phosphate increases mitochondrial membrane potential in higher eukaryotes

Given the importance of MMP for human health and disease, we tested whether phosphate depletion might also enhance MMP in the HEK293T (embryonic kidney) and A375 (melanoma) human cells. Both HEK293T and A375 cells exhibited increased MMP after 3 d of growth in phosphate-free medium (*Figure 6A*, *Figure 6—figure supplement 1A*). We quantified the morphology of the mitochondrial network based on two parameters: summed branch length mean, which describes the mean of summed length of mitochondrial tubules in each independent structure, and network branch number mean, which describes the mean number of attached mitochondrial tubules in each independent structure. Consistent with the increased MMP, HEK293T and A375 cells grown in low phosphate exhibited a more connected and elongated mitochondrial network (*Figure 6B*, *Figure 6—figure supplement 1B*). As an alternative strategy to limit phosphate uptake from the media, we treated these same cell lines with the phosphate transporter inhibitor phosphonoformic acid (PFA) for 48 hr. We observed a dose-dependent increase in MMP in both cell lines (*Figure 6C*). We concluded that phosphate depletion induces a higher MMP in cultured mammalian cell lines.

Immortalized cell lines may have adaptations that are not representative of native cells, so we decided to determine whether primary mammalian cells might also exhibit this phenomenon. Due to the special media requirements for culturing primary hepatocytes and the lack of commercially available phosphate-depleted media, we treated primary hepatocytes with PFA for 24 hr to generate

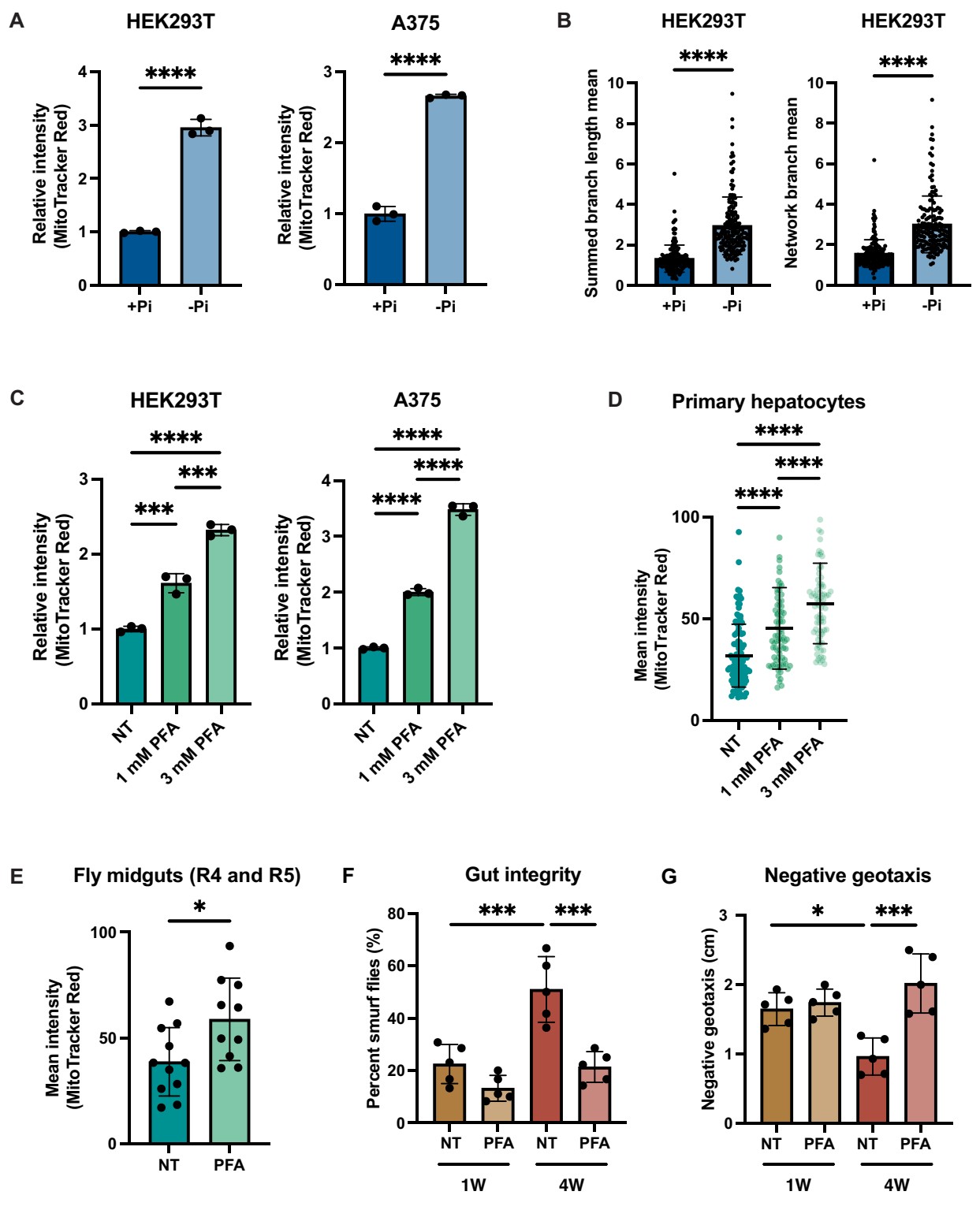

**Figure 6.** Phosphate depletion induces increased mitochondrial membrane potential in higher eukaryotic cells. (**A**) The indicated cell lines were cultured with 1 mM (+Pi) or no phosphate (-Pi) for 3 d. Mitochondrial membrane potential was quantified by flow cytometry measurement of 10,000 cells stained with MitoTracker Red. n = 3. Error bars represent the SD. Statistical significance was determined using an unpaired two-tailed *t*-test. ****p≤0.0001. (**B**) Summed branch length mean and network branch mean were measured and calculated by Mitochondrial Network Analysis (MiNA). n = 3. Error bars represent the SD. Statistical significance was determined using an unpaired two-tailed *t*-test. ****p≤0.0001. (**C**) HEK293T cells were treated with 0, 1, or 3 mM phosphonoformic acid (PFA) for 48 hr. Mitochondrial membrane potential was quantified by flow cytometry measurement

*Figure 6 continued on next page*

*Figure 6 continued*

of 10,000 cells stained with MitoTracker Red. n = 3. Error bars represent the SD. Statistical significance was determined using a one-way ANOVA with Tukey's multiple comparisons. ***$p \leq 0.0005$; ****$p \leq 0.0001$. (**D**) Primary hepatocytes were treated with 0, 1, or 3 mM PFA for 24 hr, and then stained with MitoTracker Red and imaged. The mean intensity of mitochondrial membrane potential was quantified by measurement of by the MitoTracker Red fluorescent signal from ~80 cells per condition. Error bars represent the SD. Statistical significance was determined using a one-way ANOVA with Tukey's multiple comparisons. ****$p \leq 0.0001$.(**E**) Three-week-old flies from the control group or from the experiment group treated with 1 mM PFA for 2 wk were dissected. Their midguts (R4 and R5 region) were stained with TMRE and the fluorescent signal was quantified by microscopic imaging. Error bars represent the SD. Statistical significance was determined using an unpaired two-tailed *t*-test. *$p \leq 0.05$. (**F, G**) Three-day-old flies were maintained on food either containing no drug or 1 mM PFA for 1 or 4 wk. Smurf assays were repeated for five groups, each with six female and four male flies. RING assays were repeated for five groups, each with 15 female and 10 male flies. Error bars represent the SD. Statistical significance was determined using a one-way ANOVA with Tukey's multiple comparisons. *$p \leq 0.05$; ***$p \leq 0.0005$.

The online version of this article includes the following figure supplement(s) for figure 6:

**Figure supplement 1.** Phosphate depletion induces elevated mitochondrial membrane potential in mammalian cells.

a phosphate-depleted intracellular state. Similar to the immortalized cell lines, MMP increased in a dose-dependent manner (***Figure 6D***).

Finally, we used the fruit fly *Drosophila melanogaster* to assess the relationship of phosphate starvation and MMP in an intact living animal. PFA has been previously used in flies to generate a phosphate-depleted state (***Bergwitz et al., 2013***). PFA treatment is lethal during the larval stage in the fly developmental cycle, but can extend lifespan when administered to adult flies (***Bergwitz et al., 2013***). Indeed, PFA-treated adult flies exhibited a higher MMP than the control group as measured by quantification of TMRE-stained images of the fly midgut (***Figure 6E***). We also measured the gut integrity in young and old flies treated with PFA using the Smurf assay (***Rera et al., 2012***). Older flies exhibited more Smurf dye uptake, indicative of more leakiness of the gut epithelium (***Figure 6F***). PFA treatment in either young or old flies improved gut integrity (***Figure 6F***). We also performed a negative geotaxis assay (***Gargano et al., 2005***), wherein older flies demonstrated a delayed response, which was rescued by PFA treatment (***Figure 6G***). PFA treatment increased the maximum lifespan of wild-type flies, but did not affect the average lifespan (***Figure 6—figure supplement 1C***). Altogether, these data suggest that the relationship between phosphate deprivation and MMP extends across evolution and is recapitulated in vitro and in vivo, suggesting its fundamental importance to eukaryotic biology.

## Discussion

The work described herein was initially intended to define the signaling and transcriptional network that underlies the positive and negative gene expression effects of a mutant that lacks the mtFAS system, and therefore lacks assembly of the respiratory system. This line of inquiry led to a series of observations that demonstrated a previously unappreciated role of environmental sensing and cellular signaling in controlling MMP (summarized in ***Table 1***). As a result, we propose a putative model

**Table 1.** Summary of observations in *sit4Δ* cells or cells depleted with phosphate.

| | | Perturbation | |
| --- | --- | --- | --- |
| Measurement | Background | *sit4Δ* | Phosphate depletion |
| MMP | WT | High | High |
| | *mct1Δ* | High | Higher than *mct1Δ* cells |
| | *rpo41Δ* or *rho⁰* | Higher than *rpo41Δ* cells | Higher than *rho⁰* cells |
| Mitochondrial protein import efficiency | WT | More efficient | More efficient |
| | *mct1Δ* | More efficient | More efficient than *mct1Δ* cells |
| ETC | WT | Enriched | Unchanged |
| | *mct1Δ* | More enriched than *mct1Δ* cells | Complex III, IV: remain absent Complex II: More enriched |

that, based on energetic demand, environmental status, and intracellular signaling, cells establish and maintain a MMP 'setpoint', which is tailored to maintain optimal mitochondrial function. We find that cells deploy multiple ETC-dependent and -independent strategies to maintain that setpoint. Critically, we find that cells often prioritize this MMP setpoint over other bioenergetic priorities, even in challenging environments, suggesting an important evolutionary benefit.

We first made the observation that deletion of the *SIT4* gene, which encodes the yeast homologue of the mammalian PP6 protein phosphatase, reversed many of the defects caused by loss of mtFAS, including gene expression programs, ETC complex assembly, mitochondrial morphology, and especially MMP (*Figure 1*). A previous study (*Garipler et al., 2014*) reported that the deletion of *SIT4* increased MMP in *rho⁻* cells, although the mechanism underlying the phenomenon was not defined. We show herein that increased MMP in a *sit4Δ* mutant is independent of mtDNA damage and occurs in an otherwise wild-type strain (*Figure 1C and D*). The mechanism whereby *SIT4* deletion elicits these mitochondrial effects seems to involve the induction of two gene expression programs: mitochondrial ETC biogenesis and the phosphate starvation response. Through these effects, and perhaps other mechanisms, the *sit4Δ* mutant rescues the ETC assembly failure caused by loss of mtFAS (*Figure 2B*). Indeed, the ETC complex abundance and MMP of an *mct1Δ sit4Δ* double mutant is substantially enhanced relative to a wild-type strain. This has important implications for our understanding of how mtFAS supports, but is not essential for, ETC assembly.

The transcriptional and phosphoproteomic effects of *SIT4* deletion (*Figure 3A and B*) led us to the observation that phosphate deprivation, or the perception of phosphate starvation by elimination of the Pho85-dependent phosphate sensing system, was sufficient to increase MMP. Unlike phosphate starvation, the *pho85Δ* mutant has elevated intracellular phosphate concentrations (*Gupta et al., 2019*; *Liu et al., 2017*). This suggests that the phosphate effect on MMP is likely to be elicited by cellular signaling downstream of phosphate sensing rather than some direct effect of environmental depletion of phosphate on mitochondrial energetics. However, the mechanistic details underlying the induction of phosphate starvation signaling by *SIT4* deletion remain unanswered.

One of the more surprising findings from this work is that the phosphate starvation response increases MMP independently of either of the two well-established MMP generation mechanisms, proton pumping by the ETC or ATP hydrolysis-dependent proton pumping by the ATP synthase. Phosphate starvation or the deletion of *PHO85* has minimal effects on the assembly of the respiratory complexes and does not restore detectable complexes to an *mct1Δ* strain, but still increases MMP. Phosphate deprivation even increases MMP in a mutant lacking the entire mitochondrial genome, and hence has no assembled ETC complex III or IV and no ATP synthase. Instead, our data suggest that the ADP/ATP carrier and the mitochondrial ATP hydrolysis pathway are utilized to increase MMP in response to phosphate starvation signaling (*Figure 4B and C*). The import of $ATP^{4-}$ produced by glycolysis, hydrolysis to $ADP^{3-}$ and inorganic phosphate, and export of $ADP^{3-}$ through the AAC, is an electrogenic process that is sufficient to sustain or enhance the MMP. Meanwhile, the inorganic phosphate released during ATP hydrolysis is exported with a proton through the phosphate carrier, which is an electroneutral process but contributes to the proton gradient. It was reported previously that $rho^0$ cells rely on the reverse transport of ATP and ADP through the ADP/ATP carrier in conjunction with ATP hydrolysis in the mitochondrial matrix to generate a minimal MMP (*Angerer et al., 2017*; *Buchet and Godinot, 1998*; *Chen and Clark-Walker, 2000*; *Dupont et al., 1985*; *Kovácová et al., 1968*). We now show that intracellular signaling triggers a process that can lead to an increased MMP even beyond the wild-type level in the absence of the mitochondrial genome.

The elevated MMP setpoint has significant functional consequences, as demonstrated by the partially mitochondrial Ilv2-FLAG protein becoming completely imported and cleaved upon loss of Sit4 or phosphate depletion. It is likely that many other proteins either gain or enhance their mitochondrial matrix localization and activity as well. It has been previously described that reduction in MMP can alter the import properties of proteins and thereby trigger cellular signaling events related to mitophagy, transcriptional responses, and others (*Becker et al., 2012.*; *Berry et al., 2021.*; *Jin et al., 2010*; *Miceli et al., 2011*; *Rolland et al., 2019*). The scope of responses elicited in cells experiencing high MMP, however, has not been previously interrogated to the same extent as cells with reduced MMP.

What is the evolutionary advantage of the mitochondrial ATP hydrolysis pathway, which uses the energy of ATP hydrolysis to increase the MMP? We propose a speculative hypothesis that this

mechanism enables the liberation of needed phosphate from the most abundant labile store of phosphate, ATP. However, cleaving the phosphodiester bonds of ATP releases energy in addition to releasing phosphate. Using this mitochondrial mechanism enables the cell to capture that energy in the form of the MMP rather than having it be simply lost as heat. As a result, the elevated MMP is able to fuel mitochondrial processes and empowers the cell to better combat nutrient scarcity. This phenomenon also appears to be evolutionarily conserved as cellular phosphate depletion also increases MMP in primary and immortalized human cells and in cells of the fly midgut in vivo (*Figure 6*).

Our results across the eukaryotic kingdom indicate that the higher MMP induced by phosphate deprivation contributes to improved mitochondrial energetics, morphology, and overall robustness, which could have profound implications for the many diseases, as well as the aging process itself, that are characterized by reduced MMP (*Hagen et al., 1997*; *Hughes et al., 2020*; *Leprat et al., 1990*; *Mansell et al., 2021*; *Sastre et al., 1996*; *Sugrue and Tatton, 2001*). It has been well established that phosphate limitation can extend lifespan in yeast, flies, and mice (*Bergwitz et al., 2013*; *Ebrahimi et al., 2021*; *Kurosu et al., 2005*; *Kurosu et al., 2006*). In both the fly and mouse, it was also conversely demonstrated that excessive phosphate shortens lifespan (*Bergwitz et al., 2013*; *Kuro-o et al., 1997*). In these studies, the exact mechanisms whereby phosphate abundance and sensing affect lifespan were not established. We observed increased MMP in flies treated with PFA, which limits phosphate uptake (*Figure 6E*). Other measurements showed improvement in gut integrity and negative geotaxis (*Figure 6F and G*). From the evidence presented herein (*Figure 6E–G*, *Figure 6—figure supplement 1C*), as well as the previous study showing that PFA treatment extends lifespan (*Bergwitz et al., 2013*) and enhances intestinal health in flies (*Xu et al., 2023*), we hypothesize that phosphate deprivation increased MMP and overall mitochondrial health, and thereby enabled lifespan extension. This connection is supported by a recent study that demonstrated that boosting MMP by photo-activation of an ectopically expressed proton pump is sufficient to prolong lifespan in *C. elegans* (*Berry et al., 2022*).

This work identified genetic and environmental interventions that appear to elevate the setpoint of MMP. This improves mitochondrial protein import efficiency, results in more connected mitochondrial morphology, and reverses other deficiencies found in mutants that have low MMP. This work thus proposes that MMP can be modulated through cellular signaling, including augmentation of MMP above that typically observed in a wild-type cell. It also demonstrates that this MMP augmentation can occur in the absence of a functional respiratory chain. As a result, these observations lay the foundation for future studies to identify interventions that can impact the signaling mechanisms that control MMP for therapeutic benefit.

# Materials and methods

## Yeast strains and growth conditions

*Saccharomyces cerevisiae* BY4743 (MATa/a, his3/his3, leu2/leu2, ura3/ura3, met15/MET15, lys2/LYS2) was used as a parental strain to generate all knockout strains. After using PCR-based homologous recombination method, each diploid was confirmed by genotyping the targeted region in the genome. The selected diploids were dissected after 5 d of sporulation at room temperature. The genotype of each haploid was determined by testing their corresponding auxotrophic or drug resistant ability. The genotypes of all strains used in this study are listed in *Table 2*. All plasmids, antibodies, and chemicals used in this study are listed in *Table 3*, *Table 4*, *Table 5*. Primers used to create yeast strains are listed in *Supplementary file 1*. All additional materials will be provided upon request.

Yeast transformation was performed by the LiAc/TE method, and successfully transformed cells were selected on agar plates containing synthetic complete media with 2% glucose (SD) lacking the corresponding amino acid(s). $rho^0$ cells were generated using 25 ug/ml of ethidium bromide (Sigma, E7637). Small colonies were picked to be validated with the following two assays. The $rho^0$ cells failed to grow on glycerol containing media. The isolated DNA from $rho^0$ cells yielded no product after 45 cycles of PCR with primers amplifying small fragments of *COX2* (forward: 5′-GAATGATGTACCAACACCTTATG-3′, reverse: 5′-GATACTGCTTCGATCTTAATTGGC-3′) and *ATP6* (forward: 5′-GACTATTATTTGGTTTACAATCATC-3′, reverse: 5′-TTAATGTAAGTATACTGCATCTTTTAAATATG-3′), while the DNA from wild-type cells generated robust PCR product.

**Table 2.** Yeast strains used in this study.

| Genotype | Source | JRY identifier |
|---|---|---|
| WT (BY4741) | *Van Vranken et al., 2018* | JRY 2884 |
| *mct1*::NatMX | *Van Vranken et al., 2018* | JRY 2885 |
| can1::STE2p-Sp_HIS5 lyp1del btt1::Renilla-BTT1terminator-HygMX cit2::Firefly-CIT2terminator-Met15 ho::pr-cit2-term-pr-btt1-term-ura3 mct1::NatMX | This study | JRY 4614 |
| can1::STE2p-Sp_HIS5 lyp1del btt1::Renilla-BTT1terminator-HygMX cit2::Firefly-CIT2terminator-Met15 ho::pr-cit2-term-pr-btt1-term-ura3 | This study | JRY 4616 |
| *sit4*::HygMX | This study | JRY 4144 |
| *sit4*::HygMX *mct1*::NatMX | This study | JRY 4145 |
| ILV2-FLAG::KanMX | Dr. Cory Dunn | JRY 4383/CDD1084 |
| *mct1*::NatMX ILV2-FLAG::KanMX | This study | JRY 4389 |
| *sit4*::HygMX ILV2-FLAG::KanMX | This study | JRY 4385 |
| *sit4*::HygMX *mct1*::NatMX ILV2-FLAG::KanMX | This study | JRY 4387 |
| *qcr2*::kanMX | This study | JRY 4556 |
| *qcr2*::kanMX *sit4*::HygMX | This study | JRY 4552 |
| *cox4*::kanMX | This study | JRY 4631 |
| *cox4*::kanMX *sit4*::HygMX | This study | JRY 4633 |
| *rpo41*::kanMX | This study | JRY 4719 |
| *rpo41*::kanMX *sit4*::HygMX | This study | JRY 4720 |
| *rpo41*::kanMX *sit4*::HygMX *mct1*::NatMX | This study | JRY 4721 |
| *pho85*::hisMX | This study | JRY 4715 |
| *pho85*::hisMX *mct1*::NatMX | This study | JRY 4716 |
| *pho85*::hisMX ILV2-FLAG::KanMX | This study | JRY 4743 |
| *pho85*::hisMX *mct1*::NatMX ILV2-FLAG::KanMX | This study | JRY 4744 |
| *rho$^0$* | This study | JRY 4941 |
| Tom70-yeGFP::hisMX | This study | JRY 7502 |
| Tom70-yeGFP::hisMX *mct1*::NatMX | This study | JRY 7504 |
| Tom70-yeGFP::hisMX *sit4*::HygMX | This study | JRY 7506 |
| Tom70-yeGFP::hisMX *sit4*::HygMX *mct1*::NatMX | This study | JRY 7508 |
| Tom70-yeGFP::hisMX *rho$^0$* | This study | JRY 7510 |
| Tom70-yeGFP::KanMX *mct1*::NatMX | This study | JRY 7514 |
| Tom70-yeGFP::KanMX *pho85*::hisMX | This study | JRY 7515 |
| Tom70-yeGFP::KanMX *pho85*::hisMX *mct1*::NatMX | This study | JRY 7516 |

For most assays, unless specified, yeast cells were grown in synthetic media with 2% glucose overnight and backdiluted to a much lower OD (the exact number of cells dependent on the assays and mutants being used) in media indicated in each assay. For assays that used carbon sources other than glucose, the carbon source and percentage was specified in the figure or figure legend. Cells were harvested between 0.2–0.6 OD to ensure similar metabolic state. For phosphate depletion assay, saturated or active growing yeast cells (below 0.6 OD) were washed twice with a much larger volume of water. The cells were then grown in synthetic media without inorganic phosphate supplemented

**Table 3.** Antibodies used in this study.

| Antibodies | Source |
|---|---|
| FLAG epitope | Sigma-Aldrich, F7425 |
| Sdh2 | Dr. Dennis Winge |
| Rip1 | Dr. Dennis Winge |
| Atp2 | Dr. Dennis Winge |
| Por1 | Abcam, ab110326 |
| Lipoic acid | Abcam, ab58724 |
| Pgk1 | Abcam, ab113687 |

with KCl (Formedium, CYN6701) with amino acids, 2% glucose, and the indicated amount of potassium phosphate monobasic, pH 4.1.

## Mammalian cells and growth conditions

HEK293T was obtained from ATCC. A375 was obtained from the Martin McMahon's lab. HEK293T verification was provided by ATCC. A375 was authenticated by STR profiling. Mycoplasma testing was performed once every month, and all cell lines remained mycoplasma-free. HEK293T and A375 cell lines were cultured and maintained in Dulbecco's modified Eagle medium (DMEM) supplemented with 10% FBS in an incubator at 37°C with 5% $CO_2$.

To deplete phosphate, cells were washed at minimum three times with filter-sterilized normal saline (0.9% NaCl) and trypsinized with 0.25% trypsin in citrate saline (STEMCELL Technologies, 07400). Cells were resuspended with DMEM with no phosphate (Thermo Fisher Scientific, 11971025), supplemented with 10% One Shot Dialyzed FBS (Thermo Fisher Scientific, A3382001), and 2 mM sodium pyruvate, hereby known as -Pi media, divided equally by cell number into 15 ml falcon tubes, and centrifuged to pellet cells.

Cells were resuspended in either the aforementioned -Pi media, or in +Pi media, which is media supplemented with 1 mM sodium phosphate monobasic, hereby known as +Pi media, depending on condition cells were to be plated in. Cells were grown in either -Pi or +Pi media in an incubator at 37°C and 5% $CO_2$ for 3 d prior to collection.

## Fly stocks and growth condition

*Drosophila melanogaster* stain $W^{1118}$ (BDSC 3605) was used for testing the effect of phosphate uptake on mitochondrial membrane potential. The flies were maintained on fly semi-defined food which consists of 1% agar, 8% yeast, 3% sucrose, 6% glucose, 0.05% $MgSO_4$, 0.05% $CaCl_2$, 1% Tegosept, and 0.6% propionic acid.

## Yeast genetic screen library construction and dual luciferase assay

The deletion collection (haploid) was mated with the query strain (*can1*::STE2p-Sp_HIS5 lyp1del *btt1*::Rluc-BTT1terminator-HygMX *cit2*::Fluc-CIT2terminator-Met15 *ho*:: pr-cit2-term-pr-btt1-term-ura3 *mct1*::NatMX) on YPAD plates and selected for diploid on YPAD+G418/Nat/Hyg for 2 d at 30°C. Selected diploids were sporulated on enriched sporulation media (20 g/l agar, 10 g/l potassium acetate, 1 g/l yeast extract, 0.5 g/l glucose, 0.1 g/l amino-acids supplement) for 5 d at room temperature. Desired haploids were selected on SD -his/arg/lys +canavanine/thialysine for 2 d at 30°C followed by growing on SD without ammonium sulfate supplemented with monosodium glutamic acid

**Table 4.** Plasmids used in this study.

| Plasmids | Source |
|---|---|
| pRS416 Acp1-HA-FLAG | *Van Vranken et al., 2018* |
| pRS416 Sit4-HA-FLAG | This study |

**Table 5.** Chemicals and commercial kits used in this study.

| Chemicals | Source |
| --- | --- |
| sodium phosphonoformate tribasic hexahydrate | Sigma-Aldrich, P6801 |
| b-mercaptoethanol | Sigma-Aldrich, M6250 |
| Digitonin special-grade (water-soluble) | Gold Biotechnology, D-180 |
| Lyticase from Anthrobacter luteus | Sigma-Aldrich, L4025 |
| Protease inhibitor cocktail (yeast) | Sigma-Aldrich, P8215 |
| B-ethylmaleimide | Sigma-Aldrich, E3876 |
| anti-HA antibody-conjugated agarose | Sigma-Aldrich, A2095 |
| NativePAGE 20× running buffer | Invitrogen, BN2001 |
| NativePAGE 20× cathode buffer additive | Invitrogen, BN2002 |
| NativePAGE sample buffer (4×) | Invitrogen, BN20032 |
| NativePAGE 5% G-250 sample additive | Invitrogen, BN20041 |
| Ponceau S solution | Sigma-Aldrich, P7170 |
| Antimycin A | Sigma-Aldrich, A8674 |
| Bongkretic acid | Sigma-Aldrich, B6179 |
| MitoTracker Red CMXROS | Invitrogen Life, M7512 |
| CCCP | Sigma-Aldrich, C2759 |
| TMRE | Invitrogen, T669 |
| Hoechst 33342 | Thermo Scientific, 62249 |
| Bromophenol blue | Sigma-Aldrich, 114391 |
| Pierce BCA protein assay kit | Thermo Scientific, 23225 |
| Direct-zol RNA isolation kit | Zymo Research, R2050 |
| TURBO Dnase free kit | Invitrogen Life, AM1907 |
| LightCycler 480 SYBR Green I Master | Roche Life Science, 04707516001 |
| SuperSignal West Femto Max Sensitivity Substrate | Thermo Scientific, 34096 |

(MSG) -his/arg/lys +canavanine/thialysine/G418 for 1 d at 30°C. The final selection was conducted on SD/MSG -his/arg/lys/ura/cys/met +canavanine/thialysine/G418/Nat/Hyg for 1 d at 30°C.

Haploid cells generated from SGA were grown in 384-well plates overnight in SD complete (2% glucose). After back-diluting to around OD 0.1, the cells were grown for 6 hr in SR complete media (2% raffinose). The dual luciferase assay was conducted using the Dual-Glo Luciferase Assay System (Promega) following the product manual. Both firefly and Renilla luciferase were measured using a GloMax plate reader (Promega) with an injector in 96-well plate format.

## Mitochondrial membrane potential measurement in yeast

0.2 OD of yeast cells were pelleted down and incubated in the same growth media containing 100 nM of MitoTracker Red CMXRos (Life Technologies) for 30 min at room temperature. Cells were spun down again and resuspended in the same media, which was either imaged by fluorescence microscopy or measured by fluorescence-assisted cell sorting. All experiments were performed with three biological replicates.

## MitoTracker Red staining in mammalian cells

Cells were washed 3× and trypsinized as previously stated, and then resuspended in -Pi or +Pi media with a 20 nM final concentration of MitoTracker Red, then incubated at 37°C for 15 min. Cells are

centrifuged, washed once with normal saline, centrifuged, and resuspended in -Pi or +Pi media and proceed with flow cytometry analysis. All experiments were performed with three biological replicates.

## MitoTracker Red staining and fluorescence microscopy in rat primary hepatocytes

Rat Primary hepatocytes (Wister, Lonza, RICP01) were cultured as instructed. In brief, 0.9 million primary hepatocytes were plated in collagen-coated florodish in HCM SingleQuots Kit (Lonza, CC-4182, containing ascorbic acid, bovine serum albumin-fatty acid free [BSA-FAF], hydrocorticosone, human epidermal growth factor [hEGF], transferrin, insulin and gentamicin/amphotericin-B [GA]), overnight at 37°C. Cells were treated with 3 mM or 1 mM PFA in DMEM with 10% FBS and 1% P/S for 24 hr. Cells were then washed two times with saline-0.9% NaCl before staining with 20 nM MitoTracker Red CMXRos at 37°C for 15 min. After two saline washes, live imaging of MitoTracker Red fluorescence was done in DMEM with 10% FBS on Zeiss 900 Airyscan. The intensity of MitoTracker Red for 80 cells per treatment was quantified using Fiji. Cellular boundaries were defined using the Free-hand tool, and background intensity was subtracted. The graphs were plotted using Prism v9.

## TMRE staining and fluorescence microscopy in adult *Drosophila* gut intestines

Three--day-old flies (15 females and 10 males) were transferred in vials with semi-defined media with or without 1 mM sodium phosphonoformate tribasic hexahydrate (PFA) (Sigma, P6801). After 2 wk, fly guts (midgut, R4-R5 section) from both control and PFA-treated groups were dissected in Shields and Sang M3 Insect Media (Sigma, S8398) with or without 1 mM PFA added. Dissected guts were stained with 1 µM TMRE (Sigma, 87917) in the same media condition for 20 min at room temperature. After two washes with the corresponding media supplemented with 1 µM Hoechst 33342 (Thermo Scientific, 62249) and 0.01 µM TMRE, live imaging of the gut was performed on Zeiss 900 Airyscan. Intensity of TMRE was quantified in Fiji, and background was subtracted. Figures were prepared in Prism v9.

## Fluorescence microscopy

Yeast containing Tom70-GFP tagged chromosomally at its endogenous locus were grown and stained with MitoTracker Red CMXRos as described for flow cytometry. In short, $5 \times 10^6$ yeast were harvested by centrifugation and resuspended in 1 ml media containing 0.2 µM MitoTracker Red CMXRos. Cells were incubated in the dark at room temperature for 30 min, washed once in 1 ml media, and resuspended in 20 µl of media.

For quantitative microscopy, images were collected on an Axio Observer (ZEISS) with a ×63 oil-immersion objective (ZEISS, Plan Apochromat, NA 1.4) and an Axiocam 503 mono camera (ZEISS). Three optical z-sections across 1 µm sections were collected per image. Each image contained on average 19 yeast cells (range: 5–49 yeast cells). Quantifications were derived from average values per picture. Five pictures were averaged for each condition in an experimental replicate. Final values are averages of three experimental replicates. Images were collected in ZEN (ZEISS) and processed in Fiji (*Schindelin et al., 2012*). All quantifications were done on maximum-intensity projections. All images within each experiment were processed identically.

For mitochondrial membrane potential quantifications, a mask was created from thresholded Tom70-GFP images and used to measure the average MitoTracker Red CMXRos fluorescence intensity of each mitochondria. Each condition within an experimental replicate was normalized to untreated wild-type yeast to control for variations in MitoTracker Red CMXRos staining.

For mitochondrial mass quantifications, mitochondrial area per picture was measured from thresholded Tom70-GFP images. Cell area per picture was measured from Tom70-GFP images at a threshold such that the entirety of each cell was outlined. Plots depict the total mitochondrial area divided by the corresponding cell area.

For mitochondrial morphology quantifications, the contrast of Tom70-GFP images were adjusted for optimal viewing and morphology scored by an unblinded researcher.

For image panels, super-resolution Airyscan images were collected using an LSM800 (ZEISS) equipped with an Airyscan detector and a ×63 oil-immersion objective (Carl Zeiss, Plan Apochromat, NA 1.4). Optical z-sections were acquired across the entire yeast cell at a step size optimal for Airyscan

super-resolution, 0.15 μm. Images were acquired on ZEN software and processed using the automated Airyscan processing algorithm in ZEN (ZEISS). Maximum-intensity projections and contrast enhancement were done in Fiji. Contrast changes were kept identical within an experiment.

HEK 293T imaging was performed with a Plan-Apochromat ×63/1.40 Oil DIC f/ELYR objective. Images were Airyscan processed using the ZEISS Zen Blue software.

## Negative geotaxis assay

Negative geotaxis assays were measured using the RING method (*Gargano et al., 2005*). Six females and four males of flies per experimental group were transferred to empty vials with markings every 0.5 cm along the side of the vial. Vials were tapped with force five times to ensure that all flies were at the bottom of the vial. After a recovery period of 5 s, the average height per vial that flies were able to travel up the vial was scored across all trials was scored using photos. The assay was repeated for each experimental group with five trials.

## Smurf fly assay

Smurf fly assays to report gut barrier integrity were performed as reported in *Rera et al., 2012*. Also, 15 female and 10 male 3-day-old flies per experimental group were maintained on control food or food supplemented with 1 mM PFA for 1 wk or 4 wk. These flies with reared on blue food with 1% (wt/vol) bromophenol blue (Sigma) for 1 d. The percent of Smurf flies, or flies with visible dye leaking from the abdomen, was counted for each group. The assay was repeated for each experimental group with five trials.

## Fly survival assay

In total, 15 female and 10 male 3-day-old flies per experimental group were maintained on control food or food supplemented with 1 mM PFA. For 90 d, the number of dead flies was tallied every 3 d. The assay was repeated for each experimental group with five trials.

## Fluorescence-assisted cell sorting

Cells were stained with MitoTracker Red CMXRos. A total of 10,000 events were measured on a BD FACSCanto with BD FACSDiva 8.0.1.1 (BD Biosciences). The median fluorescence values were plotted with Prism 9.

## Mitochondrial morphology quantification using MiNA

Using the MiNA plugin (*Valente et al., 2017*) in Fiji, MitoTracker Red signaling of each cell was outlined as ROI and the skeletonized mitochondria were generated. Then, the mean summed branch lengths and the mean network branch were automatically measured and calculated. Around 30–60 cells were analyzed for each condition. The final bar graphs were plotted with Prism 9.

## Mitochondrial protein import assay

Ten OD of total culture (~$10^8$ cells) were harvested at an $OD_{600}$ between 0.3 and 0.5. Cell pellets were washed with water and lysed in 500 μl of 2 M lithium acetate for 10 min on ice. Lysed cells were resuspended in 500 μl of ice-cold 0.4 M NaOH and left on ice for 10 min. The pellets were resuspended with 250 μl of 2× Laemmli buffer with 5% BME. The lysate was boiled for 5 min, and the supernatant was loaded on a 12% SDS-PAGE gel and assessed by immunoblot.

## Western blotting

Whole-cell or mitochondria extract were separated on an SDS-PAGE or BN-PAGE gel and transferred to nitrocellulose membranes (SDS-PAGE) or activated PVDF membranes (BN-PAGE) with a Power Station (Bio-Rad). Membranes were blocked in blocking buffer (Tris buffered saline [50 mM Tris–HCl pH 7.4, 150 mM NaCl, 5% nonfat dry milk]) and probed with the primary antibodies listed in *Table 3* and the secondary antibodies. Antibodies were either visualized with LI-COR Odyssey or SuperSignal Enhanced Chemiluminescence Solution (Thermo Scientific, 34096) and a Chemidoc MP System (Bio-Rad).

## Crude mitochondrial isolation

Crude mitochondrial isolation was performed as described previously (*Van Vranken et al., 2018*). Cell pellets were resuspended in TD buffer (100 mM Tris–SO$_4$, pH 9.4 and 100 mM DTT) and incubated for 15 min at 30°C. Cells were then washed once in SP buffer (1.2 M sorbitol and 20 mM potassium phosphate, pH 7.4) and incubated in SP buffer with 0.3 mg/ml lyticase (Sigma, L4025) for 1 hr at 30°C to digest the cell wall. Spheroplasts were washed once and homogenized in ice-cold SEH buffer (0.6 M sorbitol, 20 mM HEPES-KOH, pH 7.4, 1 mM PMSF, yeast protease inhibitor cocktail [Sigma, P8215]) by applying 20 strokes in a dounce homogenizer. Crude mitochondria were isolated by differential centrifugation at 3000 × *g* first to remove larger debris and 10,000 × *g* to pellet down mitochondria. Protein concentrations were determined using a Pierce BCA Protein Assay Kit (Thermo Scientific, 23225).

## Blue native polyacrylamide gel electrophoresis (BN-PAGE)

BN-PAGE was performed as described previously (*Van Vranken et al., 2018*). 100 μg of mitochondria were resuspended in 1× lysis buffer (Invitrogen, BN20032) supplemented with yeast protease inhibitor cocktail (Sigma, P8215) and solubilized with 1% digitonin for 20 min on ice. Solubilized mitochondria were cleared by centrifugation at 20,000 × *g* for 20 min. Lysate was mixed with NativePAGE 5% G-250 Sample Additive (Invitrogen, BN20041) and resolved on a 3–12% gradient native gel (Invitrogen, BN1001BOX).

## Mitochondrial isolation and immunoprecipitation for the ACP acylation study

Mitochondrial isolation and immunoprecipitation were performed as described previously (*Van Vranken et al., 2018*). Briefly, cell pellets were washed and lysed as described in the 'Crude mitochondrial isolation' section. Spheroplasts were washed once and homogenized in ice-cold SEH buffer (0.6 M sorbitol, 20 mM HEPES-KOH, pH 7.4, 1 mM PMSF, yPIC) with 10 mM N-ethylmaleimide (NEM) (Sigma, E3876) by applying 20 strokes in a dounce homogenizer. Crude mitochondria were isolated by differential centrifugation. Protein concentrations were determined using a Pierce BCA Protein Assay Kit (Thermo Scientific). 1 mg of crude mitochondria were resuspended in 200 μl of XWA buffer (20 mM HEPES, 10 mM KCl, 1.5 mM MgCl$_2$, 1 mM EDTA, 1 mM EGTA, pH 7.4) with 10 mM NEM and 0.7% digitonin added and incubated on ice for 30 min. After centrifugation at 20,000 × *g* for 20 min, solubilized mitochondria were incubated with pre-equilibrated anti-HA antibody-conjugated agarose (Sigma, A2095) for 2 hr at 4°C. The agarose was washed three times and eluted by incubating in 2× Laemmli buffer at 65°C for 10 min. Elution was isolated by SDS-PAGE and subjected for immunoblot analysis.

## RNA isolation and qPCR

RNA was isolated as described previously (*Zurita Rendón et al., 2018*). Briefly, cell pellets were resuspended in Trizol reagent (Ambion, 15596026) and bead bashed to lyse the cell (20 s bash with 30 s break on ice, six cycles). Equal volume of ethanol was added to the sample and RNA is isolated using the Direct-zol kit (Zymo Research, R2050). The RNA eluted from the column was treated with TURBO DNase kit (Invitrogen, AM1907) to remove remaining DNA contamination. After normalization of RNA content, cDNA was generated by using a High-capacity cDNA Reverse Transcription kit (Applied Biosystems, 4368813). Quantitative PCR was performed using the LightCycler 480 SYBR Green I Master (Roche, 04707516001). The raw data was analyzed by absolute quantification/second derivative of three independent biological replicates with each being the average of three technical replicates.

## RNA sequencing

For dataset GSE151606, yeast cultures were initially grown in synthetic media supplemented with 2% glucose, then removed from original media and transferred to synthetic media with 2% raffinose. Cultures were flash frozen and later total RNA was isolated using the Direct-zol kit (Zymo Research, R2050) with on-column DNase digestion and water elution. Sequencing libraries were prepared by purifying intact poly(A) RNA from total RNA samples (100–500 ng) with oligo(dT) magnetic beads and stranded mRNA sequencing libraries were prepared as described using the Illumina TruSeq Stranded

mRNA Library Preparation Kit (RS-122-2101, RS-122-2102). Sequencing libraries (25 pM) were chemically denatured and applied to an Illumina HiSeq v4 single read flow cell using an Illumina cBot. Hybridized molecules were clonally amplified and annealed to sequencing primers with reagents from an Illumina HiSeq SR Cluster Kit v4-cBot (GD-401-4001). Fifty cycle single-read sequence run was performed using HiSeq SBS Kit v4 sequencing reagents (FC-401-4002). Read pre-processing was performed using Fastp, v0.20.0 (*Chen et al., 2018*). Read alignment was performed using STAR, v2.7.3a (*Dobin et al., 2013*). Read quantification was performed using htseq, v0.11.3 (*Anders et al., 2015*). Read QC was performed using fastqc, v0.11.9 (*Andrews, 2010*). Total QC was performed using multiqc, v1.8 (*Andrews, 2010*). Library complexity QC was performed using dupradar, v1.10.0 (*Sayols et al., 2016*). Genome_build Ensembl R64-1-1 (GCA_000146045.2) version 100 was used during alignment and quantification. Genes with fewer than 10 reads in any sample were excluded from analysis. The scripts for data processing can be found at https://github.com/j-berg/ouyang_eLife2024/tree/main/rnaseq/GSE151606_mct1_timecourse (*Berg, 2024*).

For dataset GSE209726, yeast cultures were grown in SD-complete overnight and harvested at $OD_{600}$ between 0.2–0.4. Intact poly(A) RNA was purified from total RNA samples (100–500 ng) with oligo(dT) magnetic beads. Stranded mRNA sequencing libraries were prepared as described using the Illumina TruSeq Stranded mRNA Library Prep kit (20020595) and TruSeq RNA UD Indexes (20022371). Purified libraries were qualified on an Agilent Technologies 2200 TapeStation using a D1000 ScreenTape assay (Cat# 5067-5582 and 5067-5583). The molarity of adapter-modified molecules was defined by quantitative PCR using the Kapa Biosystems Kapa Library Quant Kit (Cat# KK4824). Individual libraries were normalized to 1.30 nM in preparation for Illumina sequence analysis. Sequencing libraries were chemically denatured and applied to an Illumina NovaSeq flow cell using the NovaSeq XP workflow (20043131). Following transfer of the flowcell to an Illumina NovaSeq 6000 instrument, a 150 × 150 cycle paired end sequence run was performed using a NovaSeq 6000 S4 reagent Kit v1.5 (20028312). Read preprocessing was performed using Fastp, v0.20.1 (*Chen et al., 2018*). Read alignment was performed using STAR, v2.7.7a (*Dobin et al., 2013*). Read postprocessing was performed using samtools v1.11 (*Li et al., 2009*). Read quantification was performed using htseq, v0.13.5 (*Anders et al., 2015*). Genome_build Ensembl R64-1-1 (GCA_000146045.2) version 100 was used during alignment and quantification. The scripts for data processing can be found at https://github.com/j-berg/ouyang_eLife2024/tree/main/rnaseq/GSE209726_mct1_sit4_deletions (*Berg, 2024*).

For dataset GSE212790, yeast were grown in the indicated media overnight and harvested between $OD_{600}$ = 0.2–0.4, with a total OD of 5 per sample. After QC procedures, mRNA from eukaryotic organisms is enriched from total RNA using oligo(dT) beads. The mRNA is then fragmented randomly in fragmentation buffer, followed by cDNA synthesis using random hexamers and reverse transcriptase. After first-strand synthesis, a custom second-strand synthesis buffer (Illumina) is added, with dNTPs, RNase H, and *Escherichia coli* polymerase I to generate the second strand by nick-translation and AMPure XP beads is used to purify the cDNA. The final cDNA library is ready after a round of purification, terminal repair, Atailing, ligation of sequencing adapters, size selection, and PCR enrichment. Library concentration was first quantified using a Qubit 2.0 fluorometer (Life Technologies), and then diluted to I ng/gl before checking insert size on an Agilent 2100 and quantifying to greater accuracy by quantitative PCR (Q-PCR) (library activity >2 nM). Libraries are fed into NovaSeq 6000 machines according to activity and expected data volume. A paired-end 150 bp sequencing strategy was used and all samples were sequenced to at least 6 Gb. XPRESSpipe v0.6.3 (*Berg et al., 2020*) was used to process sequence files, with the following command: xpresspipe peRNAseq... -a AGATCGGA AGAGCGTCGTGTAGGGAAAGAGTGTAGATCTCGGTGGTCGCCGTATCATT GATCGGAAGAGC ACACGTCTGAACTCCAGTCACGGATGACTATCTCGTATGCCGTCTTCTGCTTG --sjdbOverhang 149 --quantification_method htseq --remove_rrna. Genome_build Ensembl R64-1-1 (GCA_000146045.2) version 106 was used during alignment and quantification. The scripts for data processing can be found at https://github.com/j-berg/ouyang_eLife2024/tree/main/rnaseq/GSE212790_genetic_nutrient_perturbation (*Berg, 2024*).

## Data analysis and statistics for RNA sequencing

Analysis code notebooks can be accessed at https://github.com/j-berg/ouyang_eLife2024. Differential expression analysis was performed using DESeq2 (*Love et al., 2014*) with the FDR threshold (α) set at 0.1. Data visualization was performed in Python using Pandas (*McKinney, 2010*), numpy

(*Oliphant, 2006*; *van der Walt et al., 2011*), scikit-learn (*Buitinck et al., 2013*), matplotlib (*Hunter, 2007*), and seaborn (*Waskom et al., 2022*).

## Sample preparation for mass spectrometry

Yeast proteomes were extracted using a buffer containing 200 mM EPPS, 8 M urea, 0.1% SDS, and 1× protease inhibitor (Pierce protease inhibitor mini tablets). 100 µg of each proteome was prepared as follows. 10 mM tris(2-carboxyethyl)phosphine hydrochloride was incubated at room temperature for 10 min. Iodoacetimide was added to a final concentration of 10 mM to each sample and incubated for 25 min in the dark. Finally, DTT was added to each sample to a final concentration of 10 mM. A buffer exchange was carried out using a modified SP3 protocol (*Hughes et al., 2014*; *Hughes et al., 2019*). Briefly, ~500 µg of each type of SpeedBead Magnetic Carboxylate modified particles (Cytiva; 45152105050250, 65152105050250) were mixed at a 1:1 ratio and added to each sample. Then, 100% ethanol was added to each sample to achieve a final ethanol concentration of at least 50%. Samples were incubated with gentle shaking for 15 min. Samples were washed three times with 80% ethanol. Protein was eluted from SP3 beads using 200 mM EPPS pH 8.5 containing trypsin (Thermo Fisher Scientific) and Lys-C (Wako). Samples were digested overnight at 37°C with vigorous shaking. Acetonitrile was added to each sample to achieve a final concentration of 30%. Each sample was labeled in the presence of SP3 beads with ~250 µg of TMTpro-16plex reagents (Thermo Fisher Scientific) (*Li et al., 2020*; *Thompson et al., 2019*) for 1 hr. Following confirmation of satisfactory labeling (>97%), excess TMTpro reagents were quenched by addition of hydroxyl-amine to a final concentration of 0.3%. The full volume from each sample was pooled and aceto-nitrile was removed by vacuum centrifugation for 1 hr. The pooled sample was acidified using formic acid and peptides were de-salted using a Sep-Pak Vac 200 mg tC18 cartridge (Waters). Peptides were eluted in 70% acetonitrile, 1% formic acid, and dried by vacuum centrifugation. Phosphopeptides were enriched using a Hugh Select Phosphopeptide Enrichment Kit (Thermo Fisher Scientific). Flow through from the phosphopeptide enrichment column was collected for whole proteome analysis. The peptides were resuspended in 10 mM ammonium bicarbonate pH 8, 5% acetonitrile, and fractionated by basic pH reverse-phase HPLC. In total, 24 fractions were collected. The fractions were dried in a vacuum centrifuge, resuspended in 5% acetonitrile, 1% formic acid, and desalted by stage-tip. Final peptides were eluted in 70% acetonitrile, 1% formic acid, dried, and finally resuspended in 5% acetonitrile, 5% formic acid. In the end, eight fractions were analyzed by LC-MS/MS.

## Mass spectrometry data acquisition

Data were collected on an Orbitrap Eclipse mass spectrometer (Thermo Fisher Scientific) coupled to a Proxeon EASY-nLC 1000 LC pump (Thermo Fisher Scientific). Whole proteome peptides were separated using a 90 min gradient at 500 nl/min on a 30 cm column (i.d. 100 µm, Accucore, 2.6 µm, 150 Å) packed in house. High-field asymmetric-waveform ion mobility spectroscopy (FAIMS) was enabled during data acquisition with compensation voltages (CVs) set as −40 V, −60 V, and −80 V (*Schweppe et al., 2019*). MS1 data were collected using the Orbitrap (60,000 resolution; maximum injection time 50 ms; AGC $4 \times 10^5$). Determined charge states between 2 and 6 were required for sequencing, and a 60 s dynamic exclusion window was used. Data-dependent mode was set as cycle time (1 s). MS2 scans were performed in the Orbitrap with HCD fragmentation (isolation window 0.5 Da; 50,000 resolution; NCE 36%; maximum injection time 86 ms; AGC $1 \times 10^5$). Phosphopeptides were separated using a 120 min gradient at 500 nl/min on a 30 cm column (i.d. 100 µm, Accucore, 2.6 µm, 150 Å) packed in house. The phosphopeptide enrichment was injected twice using two different FAIMS methods. For the first injection, the FAIMS CVs were set to −45 V and −65 V. For the second injection, the FAIMS CVs were set to −40 V, −60 V, and −80 V (*Schweppe et al., 2019*). For both methods, MS1 data were collected using the Orbitrap (120,000 resolution; maximum ion injection time 50 ms, AGC $4 \times 10^5$). Determined charge states between 2 and 6 were required for sequencing, and a 60 s dynamic exclusion window was used. Data-dependent mode was set as cycle time (1 s). MS2 scans were performed in the Orbitrap with HCD fragmentation (isolation window 0.5 Da; 50,000 resolution; NCE 36%; maximum injection time 250 ms; AGC $1 \times 10^5$).

## Phosphoproteomics data analysis

Raw files were first converted to mzML format, and monoisotopic peaks were re-assigned using Monocle (*Rad et al., 2021*). Searches were performed using the Comet search algorithm against the most recent yeast gene database downloaded from UniProt in June 2014. We used a 50 ppm precursor ion tolerance and 0.9 Da product ion tolerance for MS2 scans collected in the ion trap and 0.02 Da product ion tolerance for MS2 scans collected in the Orbitrap. TMTpro on lysine residues and peptide N-termini (+304.2071 Da) and carbamidomethylation of cysteine residues (+57.0215 Da) were set as static modifications, while oxidation of methionine residues (+15.9949 Da) was set as a variable modification. For phosphorylated peptide analysis, +79.9663 Da was set as a variable modification on serine, threonine, and tyrosine residues.

Peptide-spectrum matches (PSMs) were adjusted to a 1% false discovery rate (FDR) (*Elias and Gygi, 2007*). PSM filtering was performed using linear discriminant analysis (LDA) as described previously (*Huttlin et al., 2010*), while considering the following parameters: comet log expect, different sequence delta comet log expect (percent difference between the first hit and the next hit with a different peptide sequence), missed cleavages, peptide length, charge state, precursor mass accuracy, and fraction of ions matched. Each run was filtered separately. Protein-level FDR was subsequently estimated at a data set level. For each protein across all samples, the posterior probabilities reported by the LDA model for each peptide were multiplied to give a protein-level probability estimate. Using the Picked FDR method (*Savitski et al., 2015*), proteins were filtered to the target 1% FDR level. Phosphorylation site localization was determined using the AScore algorithm (*Beausoleil et al., 2006*).

For reporter ion quantification, a 0.003 Da window around the theoretical *m/z* of each reporter ion was scanned, and the most intense *m/z* was used. Reporter ion intensities were adjusted to correct for the isotopic impurities of the different TMTpro reagents according to the manufacturer's specifications. Peptides were filtered to include only those with a summed signal-to-noise (SN) of 160 or greater across all channels. For each protein, the filtered peptide TMTpro SN values were summed to generate protein quantification.

## Acknowledgements

We thank the University of Utah core facilities, especially James Marvin, PhD, at the Flow Cytometry Core, Brian Dalley, PhD, at the High-Throughput Genomics Core, and the DNA/Peptide Synthesis Core. We thank members of the Rutter lab for discussion and feedback on the manuscript. Several of the figures were created with BioRender.com. This study was supported by 1F32GM140525 to CNC; 1T32DK11096601 and 1F99CA253744 to JAB; 1F30CA243440-01A1 to JMW; 1K99HL168312-01 to AAC; R01GM110755 to DRW; R35GM131854 to JR. JGV is the Mark Foundation for Cancer Research Fellow of the Damon Runyon Cancer Research Foundation (DRG-2359-19). JR is an Investigator of the Howard Hughes Medical Institute.

## Additional information

### Funding

| Funder | Grant reference number | Author |
|---|---|---|
| National Institutes of Health | 1F32GM140525 | Corey N Cunningham |
| National Institutes of Health | 1T32DK11096601 | Jordan A Berg |
| National Institutes of Health | 1F99CA253744 | Jordan A Berg |
| National Institutes of Health | 1F30CA243440-01A1 | Jacob M Winter |
| National Institutes of Health | 1K99HL168312-01 | Ahmad A Cluntun |

| Funder | Grant reference number | Author |
|---|---|---|
| National Institutes of Health | R01GM110755 | Dennis R Winge |
| National Institutes of Health | R35GM131854 | Jared Rutter |
| Damon Runyon Cancer Research Foundation | DRG-2359-19 | Jonathan G Van Vranken |
| Howard Hughes Medical Institute | | Jared Rutter |

The funders had no role in study design, data collection and interpretation, or the decision to submit the work for publication.

### Author contributions

Yeyun Ouyang, Conceptualization, Data curation, Formal analysis, Validation, Investigation, Visualization, Methodology, Writing – original draft, Writing – review and editing; Mi-Young Jeong, Ahmad A Cluntun, Geanette Lam, Investigation, Methodology; Corey N Cunningham, Validation, Investigation, Methodology, Writing – review and editing; Jordan A Berg, Data curation, Software, Formal analysis, Funding acquisition, Visualization, Writing – review and editing; Ashish G Toshniwal, Formal analysis, Validation, Investigation, Visualization, Methodology, Writing – review and editing; Casey E Hughes, Conceptualization, Formal analysis, Validation, Investigation, Visualization, Methodology, Writing – review and editing; Kristina Seiler, Katja K Dove, Greg Odorizzi, Investigation; Jonathan G Van Vranken, Data curation, Software, Formal analysis, Funding acquisition, Methodology; Jacob M Winter, Methodology; Emel Akdogan, Steven P Gygi, Cory D Dunn, Resources; Sara M Nowinski, Conceptualization; Matthew West, Formal analysis, Investigation; Dennis R Winge, Funding acquisition; Jared Rutter, Conceptualization, Resources, Supervision, Funding acquisition, Writing – original draft, Project administration, Writing – review and editing

### Author ORCIDs

Yeyun Ouyang ![ORCID] http://orcid.org/0000-0001-9523-1044
Jordan A Berg ![ORCID] http://orcid.org/0000-0002-5096-0558
Jonathan G Van Vranken ![ORCID] http://orcid.org/0000-0002-8931-852X
Ahmad A Cluntun ![ORCID] http://orcid.org/0000-0001-7612-8375
Greg Odorizzi ![ORCID] http://orcid.org/0000-0002-1143-1098
Steven P Gygi ![ORCID] http://orcid.org/0000-0001-7626-0034
Dennis R Winge ![ORCID] http://orcid.org/0000-0003-1160-1189
Jared Rutter ![ORCID] https://orcid.org/0000-0002-2710-9765

### Decision letter and Author response

Decision letter https://doi.org/10.7554/eLife.84282.sa1
Author response https://doi.org/10.7554/eLife.84282.sa2

## Additional files

### Supplementary files

• Supplementary file 1. Primers used to create yeast strains.
• Supplementary file 2. RNA-sequencing results.
• Supplementary file 3. Phosphoproteomics results.
• MDAR checklist

### Data availability

The mass spectrometry data have been deposited to the ProteomeXchange Consortium with the data set identifier PXD037405. RNA sequencing data have been deposited to the GEO Omnibus Repository with data set identifiers GSE151606, GSE212790, and GSE209726. Code for high-throughput dataset analysis is available on GitHub (https://github.com/j-berg/ouyang_eLife2024; copy archived at *Berg, 2024*) under an MIT license.

The following datasets were generated:

| Author(s) | Year | Dataset title | Dataset URL | Database and Identifier |
|---|---|---|---|---|
| Van Vranken JG, Ouyang Y, Rutter J | 2024 | Phosphate Starvation Signaling Increases Mitochondrial Membrane Potential through Respiration-independent Mechanisms | http://proteomecentral.proteomexchange.org/cgi/GetDataset?ID=PXD037405 | ProteomeXchange, PXD037405 |
| Ouyang Y, Berg JA, Rutter J | 2022 | Sequencing of yeast mutants with or without phosphate depletion | http://www.ncbi.nlm.nih.gov/geo/query/acc.cgi?acc=GSE212790 | NCBI Gene Expression Omnibus, GSE212790 |
| Ouyang Y, Berg JA, Rutter J | 2022 | Sequencing of yeast mutants | http://www.ncbi.nlm.nih.gov/geo/query/acc.cgi?acc=GSE209726 | NCBI Gene Expression Omnibus, GSE209726 |

The following previously published dataset was used:

| Author(s) | Year | Dataset title | Dataset URL | Database and Identifier |
|---|---|---|---|---|
| Nowinksi SM, Berg JA, Rutter J | 2020 | MCT1 deletion in *Saccharomyces cerevisiae* | https://www.ncbi.nlm.nih.gov/geo/query/acc.cgi?acc=GSE151606 | NCBI Gene Expression Omnibus, GSE151606 |

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
