## [Editor Report]

Mitochondrial inner membrane potential is a key factor determining several mitochondrial functions, i.e. respiration and protein import, and, thus, affects cellular metabolism. The study identifies a novel mechanism involving phosphate regulation involved in enhancement of inner membrane potential. These fundamental findings are supported by compelling evidence, with rigorous biochemical and state-of-the-art methodology. The results contribute to basic biology knowledge but also open possibilities to modulate mitochondrial potential for therapeutic purposes.

---

## [Decision Letter]

**Decision letter after peer review:**

Thank you for submitting your article "Phosphate Starvation Signaling Increases Mitochondrial Membrane Potential through Respiration-independent Mechanisms" for consideration by *eLife*. Your article has been reviewed by 3 peer reviewers, and the evaluation has been overseen by a Reviewing Editor and Benoît Kornmann as the Senior Editor. The following individuals involved in review of your submission have agreed to reveal their identity: Johannes M Herrmann (Reviewer #1); Jean-Paul di Rago (Reviewer #2).

The reviewers have discussed their reviews with one another, and the Reviewing Editor has drafted this to help you prepare a revised submission. There are a number of issues that are necessary to improve this interesting study with the most important ones specified below and in the individual comment section of the reviewers.

Essential revisions:

1) The study shows only data on glucose. Thus the role of carbon sources, and the HAP complex, needs to be considered.

2) A method, independent from mitotracker, should be included to measure membrane potential.

3) The role of amino acid metabolism should be clarified due to its central relevance for this study.

4) The authors should discuss the early literature on the petite negativity and the phosphate cycle.

*Reviewer #1 (Recommendations for the authors):*

1. Page 7: ‚as do cell experiencing amino acid starvation' should read 'as do cells experiencing amino acid starvation'.

2. Figure 1G shows the quantification of Ilv2 precursors relative to Pgk1. It would be better to compare the signal of the precursor to that of the mature Ilv2-FLAG protein. The result would be presumably similar but since the overall Ilv2 levels might vary, a comparison of the precursor vs. total Ilv2 seems more appropriate.

*Reviewer #2 (Recommendations for the authors):*

1) page 15: "Whether or not complex III or IV was inactivated, deletion of SIT4 was sufficient to increase MMP (Figure 2D) ".

This statement is partially incorrect. Indeed, in the absence of QCR2 or Cox2, about 80% of the MPP increase induced by the loss of Sit4p is lost (Figure 2D).

2) Page 16: "These results demonstrated that although ETC activity is required for the majority of the enhanced membrane potential observed in sit4D cells, sit4D mutants clearly leverage additional ETC- and ATP synthase-independent mechanisms to increase mitochondrial membrane potential (Figure 2F)".

This conclusion is premature (and possibly not valid) because cells lacking functional mtDNA need the F1 component of ATP synthase to maintain a sufficient MMP (in combination with Aac2) and hence viability.

3) page 16: "Among the most enriched phosphoproteins, Pho84, Vtc3, and Spl2, are all involved in the regulation of intracellular phosphate levels (Figure 3A, Supplementary File 3)."

What about Mir1, the protein responsible for the import of phosphate into the mitochondrial matrix?

4) page 18: "However, depleting phosphate in rho0 cells failed to rescue the absence of the complexes".

This comment looks a bit ridiculous (the petite mutation is irreversible and without mtDNA Complexes III, IV and V cannot be synthesized at all).

5) Page 19: "Importantly, combined treatment with both antimycin A and bongkrekic acid completely blocked the induction of MMP in response to low phosphate (Figure 4B). As a genetic alternative to antimycin A inhibition of the ETC, we grew rho0 cells, which have no complex III and IV nor complete ATP synthase, in low and high phosphate (Figure 4C). As shown before, phosphate depletion triggers an enhanced MMP in rho0 cells, but this is completely eliminated by bongkrekic acid in a dose-dependent manner (Figure 4C)."

The authors have missed several previously published studies where the effects of these inhibitors on the MMP have been described. Would they want to keep these data, the authors should indicate that this was already observed previously and quote the corresponding papers.

6). Page 19: "These experiments suggest a mechanism whereby the depletion of phosphate increases MMP in an ETC- and ATP synthase-independent manner".

As above (point 2), the authors should consider the possible involvement of F1 in the modulation of MMP by phosphate availability.

7) Page 20: "When the ADP/ATP carrier imports ATP4- and exports ADP3-, a net export of a positive charge occurs out of the matrix to the inter-membrane space (Figure 4D). This activity must be coupled to ATP hydrolysis within the mitochondrial matrix by an as yet unidentified ATPase.

True but this ATPase has been identified (Giraud and Velours, EJB 1994, Lefebvre-Legendre et al., Mol. Microbiol. 2003): it is the F1 component of ATP synthase.

8). Page 20: "Our data suggest that when cells lack the proton pumping ability of the ETC--either by chemical (treatment with antimycin A) or genetic (loss of mtDNA in rho0 cells) inhibition--and particularly during phosphate depletion, they instead rely on the ADP/ATP carrier to increase MMP to sustain critical mitochondrial functions".

Again, this is known and has been described already (see point 7).

9) Page 26: "It was reported previously that rho0 cells rely on the reverse transport of ATP and ADP through the ADP/ATP carrier in conjunction with ATP hydrolysis in the mitochondrial matrix to generate a minimal MMP (Appleby et al., 1999; Buchet and Godinot, 1998; X. J. Chen and Clark-Walker, 2000; Dupont et al., 1985; Kov.čov. et al., 1968)."

The authors should add the two references showing that it is the F1.ATPAse that works in conjunction with Acc2 to maintain a sufficient MMP in cells that lack functional mtDNA or unable to respire (Giraud and Velours, EJB 1994, Lefebvre-Legendre et al., Mol. Microbiol. 2003).

*Reviewer #3 (Recommendations for the authors):*

The choice to focus on SIT4 after the screen has not fully been justified. If the authors have done a screen for gene expression of their 17 validated hits for expression of the ETC genes QCR2 and RIP1, then they ought to show that data for all the hits, rather than just SIT2. In their conclusion the authors mention that

"A previous study (Garipler et al., 2014) reported that the deletion of SIT4 increased MMP in rho- cells".

If that is the case, perhaps this should be mentioned as part of the rationale for focusing on SIT4.

The authors state

"Consistent with a previous report (Jablonka et al., 2006), sit4D cells failed to grow on media containing a non-fermentable carbon source such as glycerol that requires mitochondrial respiration (Figure S2C). This defect was rescued by re-expression of SIT4 on a plasmid, confirming that sit4D cells have functional mtDNA."

It is unclear to me how reexpression on a plasmid shows anything about whether or not the mtDNA are functional without Sit4p. Surely if Sit4p is expressed on a plasmid, growth in that strain can say nothing about whether mtDNA is functional without Sit4p. The strain with SIT4 expression from a plasmid in a sit4del background does not appear to be listed in the list of strains used in the study.

It is the case with any scientific inquiry that some questions will be answered and some new questions will arise. I would have appreciated the discussion more if some of these remaining unanswered questions were highlighted. In particular the link between SIT4 deletion and phosphate starvation signaling is still unresolved; SIT4 deletion rescues MMP and ilv2p import defects in an mct1D background while phosphate starvation does not. Linking this discussion to a schematic summarizing the study's findings related to phosphate starvation, SIT4 and the genes of the ETC (perhaps as a final panel in the last figure) might also be helpful.

Figure S1A:

The caption should indicate that this is a reanalysis of previously published data and reference the paper.

Figure S1A, the authors state that deletion of MCT1 deletion demonstrates induction of mRNAs encoding subunits of ETC and ATP Synthase. It is hard to assess these observations just by looking at the gene names. It is not clear how genes were selected for figure S1A. For instance, COB and QCR9 seem to be missing from the complex III genes and one of the cytochrome C isoforms, CYC7 is missing. It would be more convincing if the statement was quantified – a percentage of the genes related to the ETC and ATP synthase (e.g. from the go term, yeast pathway annotation, or a list from a reference) that meet some threshold of induction in WT cells but not in mct1Del cells. A similar point can be made for genes related to acetyl-CoA production in Figure S1B.

Also in both those figures, the legend was hard to read and the caption did not explain the experiment (i.e. that the culture was switched from glucose to raffinose at Time 0). For figure S1B the blue, red, and green text was not explained in the caption.

The validation S1D would have been more convincing if it had included RTQPCR with CIT2.

For Figure 1A – it seemed like expression of Yat2 in the independently generated Sit4 mutant was missing. I assume the authors were unable to collect the data for some technical reason, but one would expect a change in Yat 2 as well as it was one of the selection criteria.

For Figure 1C it would be useful to show the expression of CIT2 alongside BTT1 in the gene expression data to help introduce and illustrate the screen.

A supplementary figure with a scatterplot of the data from the dual luciferace screen indicating the ratio used as a cutoff to select hits would give more confidence in the data. Also it is not clear if replicates were done for the screen and what cutoffs were used to define reduced expression. I assume the authors mean reduced expression at some time point after switching from Glucose to Raffinose relative to the parental strain with the dual luciferase construct, but cannot be sure. In particular the caption of Figure 1D is confusing.

For Fig1D, the captions says that normalized mitochondrial membrane potential is plotted. It is not clear from the methods how the flow cytometry measurements were normalized. Were they normalized to WT in each experiment like the microscope, or to Side Scatter or Forward Scatter which is related to cell size and which is often correlated with intensity in flow cytometry.

The conditions under which mRNA abundance was measured for the sit4del and WT strain for Figure 2A (SC overnight) ought to be mentioned in the caption (and possibly in the text) especially as these conditions differ from the Raffinose conditions that the previous RNA-seq experiments were performed in.

On Page 14: "As expected, the mct1D sit4D double mutant also failed to grow under respiratory conditions (Figure S2D).", refers to the wrong figure, it should refer to S2C.

For Figure 2C, the authors state "'Moreover, the observed modest increase in oxygen consumption is insufficient to explain the profound increase in MMP observed in the mct1D sit4D double mutant."

This is confusing wording because there is not an increase in Oxygen Consumption relative to WT in the mct1del sit4del double mutant, but rather a decrease. I assume the authors mean increase with respect to the mct1del mutant.

In Figure S2F it would be helpful to label the F0 component of the ATP synthase if that is what disappears in the rpo41del strains. I was a bit confused because the antibody for Atp2p, part of the F1 component was shown.

Figure 2F is a bit hard to interpret because the authors suggest that the mitochondrial membrane potential is not increased by reversal of the ATP synthase. After reading through the relevant text it seems like what the authors want to say is that they have ruled out the reversal of ATP synthase as a driver of the increased MMP in sit4del yeast. The dark arrow towards Mitochondrial Membrane potential makes it look like reversal of the ATP synthase is contributing to the increase of mitochondrial membrane potential. Perhaps it would be easier to interpret if there was a red cross below reversal of the ATP Synthase as well?

Figure 4B shows fold increases with and without Phosphate – that would also be useful to show on other such comparisons (1A, 1B, 2C, 2D).

Figure 4C. Caption does not list both drugs.

Figure 4D: It would be helpful to show BKA on this schematic.

While deletion of PHO85 in combination with deletion of MCT1 helps to establish a link between phosphate starvation and mitochondrial integrity (FiguresS4B-D) it would be more direct to test phosphate starvation directly in an mct1del background.

The phrase "The scope of responses elicited in cells experiencing high MMP, however, has not been previously interrogated." is hard to evaluate. It seems from the authors previous sentence that a lot has already been done to interrogate the scope of responses to high MMP. It is also not obvious that this study fully interrogates the scope of responses. This statement may need to be qualified or made more clear.

In the methods section for the Mitochondrial Protein Import Assay it is unclear what '10OD of total culture at an OD600 between 0.3 and 0.5' means. Approximate total number of cells or a volume should be provided.

---

## [Author Response]

Essential revisions:Reviewer #1 (Recommendations for the authors):1. Page 7: ‚as do cell experiencing amino acid starvation' should read 'as do cells experiencing amino acid starvation'.

We appreciate this catch by the reviewer and have corrected this typo in the updated manuscript.

2. Figure 1G shows the quantification of Ilv2 precursors relative to Pgk1. It would be better to compare the signal of the precursor to that of the mature Ilv2-FLAG protein. The result would be presumably similar but since the overall Ilv2 levels might vary, a comparison of the precursor vs. total Ilv2 seems more appropriate.

We agree with the reviewer that this is a valuable comparison and have performed all of the quantifications suggested. We now use unimported over total Ilv2-FLAG for all quantifications.

Reviewer #2 (Recommendations for the authors):1) page 15: "Whether or not complex III or IV was inactivated, deletion of SIT4 was sufficient to increase MMP (Figure 2D) ".This statement is partially incorrect. Indeed, in the absence of QCR2 or Cox2, about 80% of the MPP increase induced by the loss of Sit4p is lost (Figure 2D).

We appreciate this suggestion from the reviewer and have thus rephrased the sentence to ensure accuracy. It now reads "Whether or not complex III or IV was inactivated, deletion of *SIT4* was sufficient to increase MMP albeit not to the same extent as in wildtype cells (Figure 2D) ".

2) Page 16: "These results demonstrated that although ETC activity is required for the majority of the enhanced membrane potential observed in sit4D cells, sit4D mutants clearly leverage additional ETC- and ATP synthase-independent mechanisms to increase mitochondrial membrane potential (Figure 2F)".This conclusion is premature (and possibly not valid) because cells lacking functional mtDNA need the F1 component of ATP synthase to maintain a sufficient MMP (in combination with Aac2) and hence viability.

A similar concern was raised by reviewer #1 – major concern #3. We have addressed this concern in detail with new data (Figure 5—figure supplement 1F-G) incorporated in the main text and refer the reviewer to the comments above in this document.

3) page 16: "Among the most enriched phosphoproteins, Pho84, Vtc3, and Spl2, are all involved in the regulation of intracellular phosphate levels (Figure 3A, Supplementary File 3)."What about Mir1, the protein responsible for the import of phosphate into the mitochondrial matrix?

We looked through our phosphoproteomics data, and unfortunately no Mir1 peptides were detected in this experiment. We thus do not feel comfortable speculating as to Mir1’s role in this context.

4) page 18: "However, depleting phosphate in rho0 cells failed to rescue the absence of the complexes".This comment looks a bit ridiculous (the petite mutation is irreversible and without mtDNA Complexes III, IV and V cannot be synthesized at all).

We appreciate this suggestion and rephrased this sentence to read "However, as expected, depleting phosphate in *rho^0^* cells failed to rescue the absence of the complexes".

5) Page 19: "Importantly, combined treatment with both antimycin A and bongkrekic acid completely blocked the induction of MMP in response to low phosphate (Figure 4B). As a genetic alternative to antimycin A inhibition of the ETC, we grew rho0 cells, which have no complex III and IV nor complete ATP synthase, in low and high phosphate (Figure 4C). As shown before, phosphate depletion triggers an enhanced MMP in rho0 cells, but this is completely eliminated by bongkrekic acid in a dose-dependent manner (Figure 4C)."The authors have missed several previously published studies where the effects of these inhibitors on the MMP have been described. Would they want to keep these data, the authors should indicate that this was already observed previously and quote the corresponding papers.

With the initial submission, we decided not to elaborate on the historical context of this mechanism in the result section but rather to keep this section concise and focused on the data and data interpretation. In the Discussion section, we cited and discussed extensively all relevant literature and stated what our work expands upon these previous discoveries and models.

To address another concern raised by multiple reviewers, we added a few sentences and relevant citations regarding the ADP/ATP carrier phenomenon in the Results section in this revised manuscript (page 23).

6). Page 19: "These experiments suggest a mechanism whereby the depletion of phosphate increases MMP in an ETC- and ATP synthase-independent manner".As above (point 2), the authors should consider the possible involvement of F1 in the modulation of MMP by phosphate availability.

This is a similar comment as raised by reviewer #1 – major concern #3, which we have addressed above with a discussion and additional data. Briefly, we do not think the F_1_ subunit is required for the ATP hydrolysis activity to generate MMP in situations with phosphate depletion. We believe there are additional ATPase(s) in the mitochondrial matrix that can be utilized to couple the ADP/ATP carrier to MMP generation during phosphate starvation. This discussion and relevant data (Figure 5—figure supplement 1F-G) are now included in the revised manuscript.

7) Page 20: "When the ADP/ATP carrier imports ATP4- and exports ADP3-, a net export of a positive charge occurs out of the matrix to the inter-membrane space (Figure 4D). This activity must be coupled to ATP hydrolysis within the mitochondrial matrix by an as yet unidentified ATPase.True but this ATPase has been identified (Giraud and Velours, EJB 1994, Lefebvre-Legendre et al., Mol. Microbiol. 2003): it is the F1 component of ATP synthase.

Please refer to the comments above (concern #6). We included a paragraph with more detailed discussion in the main text on page 23.

8). Page 20: "Our data suggest that when cells lack the proton pumping ability of the ETC--either by chemical (treatment with antimycin A) or genetic (loss of mtDNA in rho0 cells) inhibition--and particularly during phosphate depletion, they instead rely on the ADP/ATP carrier to increase MMP to sustain critical mitochondrial functions".Again, this is known and has been described already (see point 7).

We would like to follow the same principle as stated in the response to concern #5. We fully acknowledged the previous publications and extensively discussed the relevance in the Discussion section with our initial submission. We now added relevant citations in the result section as well.

9) Page 26: "It was reported previously that rho0 cells rely on the reverse transport of ATP and ADP through the ADP/ATP carrier in conjunction with ATP hydrolysis in the mitochondrial matrix to generate a minimal MMP (Appleby et al., 1999; Buchet and Godinot, 1998; X. J. Chen and Clark-Walker, 2000; Dupont et al., 1985; Kov.čov. et al., 1968)."The authors should add the two references showing that it is the F1.ATPAse that works in conjunction with Acc2 to maintain a sufficient MMP in cells that lack functional mtDNA or unable to respire (Giraud and Velours, EJB 1994, Lefebvre-Legendre et al., Mol. Microbiol. 2003).

We think there are additional ATPase(s) other than the F_1_ subunit of ATPase that hydrolyze ATP induced and are induced by phosphate depletion. We provide an updated discussion (see page 23) and new data (Figure 5—figure supplement 1F-G).

Reviewer #3 (Recommendations for the authors):The choice to focus on SIT4 after the screen has not fully been justified. If the authors have done a screen for gene expression of their 17 validated hits for expression of the ETC genes QCR2 and RIP1, then they ought to show that data for all the hits, rather than just SIT2. In their conclusion the authors mention that"A previous study (Garipler et al., 2014) reported that the deletion of SIT4 increased MMP in rho- cells".If that is the case, perhaps this should be mentioned as part of the rationale for focusing on SIT4.

We appreciate this concern and have rewritten the relevant rationale for following up with *SIT4*. The sentence now reads “Due to the gene expression data and previous literature establishing a role of *SIT4* in regulating OXPHOS, we sought to understand how…”.

The authors state"Consistent with a previous report (Jablonka et al., 2006), sit4D cells failed to grow on media containing a non-fermentable carbon source such as glycerol that requires mitochondrial respiration (Figure S2C). This defect was rescued by re-expression of SIT4 on a plasmid, confirming that sit4D cells have functional mtDNA."

We appreciate this concern and have rephased this sentence to avoid misunderstanding or misinterpretation. This sentence now reads “This defect was rescued by re-expression of *SIT4* on a plasmid, confirming that *sit4*D cells do not have an irreversible loss of mtDNA as would be observed in a *rho^0^* stain.” We also included the plasmid containing Sit4-HA-FLAG in our material table. We hope this is more clear and prevents any confusion.

It is unclear to me how reexpression on a plasmid shows anything about whether or not the mtDNA are functional without Sit4p. Surely if Sit4p is expressed on a plasmid, growth in that strain can say nothing about whether mtDNA is functional without Sit4p. The strain with SIT4 expression from a plasmid in a sit4del background does not appear to be listed in the list of strains used in the study.It is the case with any scientific inquiry that some questions will be answered and some new questions will arise. I would have appreciated the discussion more if some of these remaining unanswered questions were highlighted. In particular the link between SIT4 deletion and phosphate starvation signaling is still unresolved; SIT4 deletion rescues MMP and ilv2p import defects in an mct1D background while phosphate starvation does not. Linking this discussion to a schematic summarizing the study's findings related to phosphate starvation, SIT4 and the genes of the ETC (perhaps as a final panel in the last figure) might also be helpful.

We appreciate this perspective, and wholeheartedly agree that there are many similarities and differences between *sit4*D cells and cells starved with phosphate. To help clarify these points, we have included a Table 5 that summarizes our observations. We have also included unanswered link between *SIT4* deletion and phosphate starvation signals in the Discussion section (page 26).

Figure S1A:The caption should indicate that this is a reanalysis of previously published data and reference the paper.

We have rephrased the caption for Figure 1—figure supplement 1A to reiterate that these two heatmaps were generated by reanalyzing published data. The caption now reads “Heat map visualizing selected gene expression between wild-type (WT) and *mct1*D using transcriptomics data from Berg et al., 2023.”

Figure S1A, the authors state that deletion of MCT1 deletion demonstrates induction of mRNAs encoding subunits of ETC and ATP Synthase. It is hard to assess these observations just by looking at the gene names. It is not clear how genes were selected for figure S1A. For instance, COB and QCR9 seem to be missing from the complex III genes and one of the cytochrome C isoforms, CYC7 is missing. It would be more convincing if the statement was quantified – a percentage of the genes related to the ETC and ATP synthase (e.g. from the go term, yeast pathway annotation, or a list from a reference) that meet some threshold of induction in WT cells but not in mct1Del cells. A similar point can be made for genes related to acetyl-CoA production in Figure S1B.

Genes were excluded from analysis if any sample contained fewer than 10 reads, as is commonplace in sequencing analysis (see https://training.galaxyproject.org/trainingmaterial/topics/transcriptomics/tutorials/rna-seq-counts-to-genes/tutorial.html#filteringto-remove-lowly-expressed-genes for a good explanation behind this rationale). As a result, *CYT1*, *COB*, *QCR9*, *CYC7*, *COX1-3*, *ATP6*, *ATP8*, and *OLI1* were filtered out during the analysis. We now added this clarification in the method section.

Also in both those figures, the legend was hard to read and the caption did not explain the experiment (i.e. that the culture was switched from glucose to raffinose at Time 0). For figure S1B the blue, red, and green text was not explained in the caption.

We have rewritten the Figure 1—figure supplement 1A-B legend to include more experimental details as suggested, as well as read more clearly. Genes in red and blue font in heat maps were followed up in the qPCR experiment shown in Figure 1A-B.

The validation S1D would have been more convincing if it had included RTQPCR with CIT2

During the query strain construction for SGA, we swapped the original *CIT2* locus with firefly luciferase. Even though we re-expressed *CIT2* at *HO* locus, the *CIT2* expression does not accurately reflect what it would be in a more wild-type strain. Therefore, we chose other genes that were also upregulated in *mct1*D cells as a proxy.

For Figure 1A – it seemed like expression of Yat2 in the independently generated Sit4 mutant was missing. I assume the authors were unable to collect the data for some technical reason, but one would expect a change in Yat 2 as well as it was one of the selection criteria.

It is not a technical reason but simply that we want to keep four target genes. When we made *sit4*D cells in-house without all the luciferase construct in the genome, we could measure *CIT2* expression. We think *CIT2* expression is a more meaningful data point than *YAT1* because it is to what the original screen readout used was (in the form of firefly luciferase). Therefore, we did not include *YAT1* in the qPCR experiment.

For Figure 1C it would be useful to show the expression of CIT2 alongside BTT1 in the gene expression data to help introduce and illustrate the screen.

Both *BTT1* and *MRL2* expression do not change between perturbations according to the RNA-seq data. Thus, for practical reasons we chose one over the other for different experiments.

A supplementary figure with a scatterplot of the data from the dual luciferace screen indicating the ratio used as a cutoff to select hits would give more confidence in the data. Also it is not clear if replicates were done for the screen and what cutoffs were used to define reduced expression. I assume the authors mean reduced expression at some time point after switching from Glucose to Raffinose relative to the parental strain with the dual luciferase construct, but cannot be sure. In particular the caption of Figure 1D is confusing.

We do not have the ability to show this data as a meaningful and interpretable scatterplot. The initial screen had a lot of plate-to-plate variability, which makes it hard to calculate a threshold for each plate. In addition, each mutant had different growth rates and reached different ODs before the luciferase assay, which further complicates the interpretation of data and the interpolate normalization required to display all of this screen in one succinct scatterplot. However, it is important to note that because of this plate-to-plate variability, we performed an extensive secondary screen and validated all the hits. We picked promising hits from the first round to re-run the dual luciferase assay in a low throughput and more precise manner, which allowed for better control for the OD at harvest. We now include these details in the updated manuscript.

For Fig1D, the captions says that normalized mitochondrial membrane potential is plotted. It is not clear from the methods how the flow cytometry measurements were normalized. Were they normalized to WT in each experiment like the microscope, or to Side Scatter or Forward Scatter which is related to cell size and which is often correlated with intensity in flow cytometry.

We used the median fluorescent intensity of 10,000 cells measured by flow cytometer. We analyzed and plotted three biological replicates of each strain with the average fluorescent intensity of wild-type cells set to 1.

The conditions under which mRNA abundance was measured for the sit4del and WT strain for Figure 2A (SC overnight) ought to be mentioned in the caption (and possibly in the text) especially as these conditions differ from the Raffinose conditions that the previous RNA-seq experiments were performed in.

We added this detail in the Figure 2 legend as suggested. The sentence now reads “Volcano plot of the transcriptomics data of *sit4*D vs. wild-type (WT) cells grown in synthetic media containing 2% glucose.”

On Page 14: "As expected, the mct1D sit4D double mutant also failed to grow under respiratory conditions (Figure S2D).", refers to the wrong figure, it should refer to S2C.

We appreciate this observation and have corrected this mistake in the updated manuscript.

For Figure 2C, the authors state "'Moreover, the observed modest increase in oxygen consumption is insufficient to explain the profound increase in MMP observed in the mct1D sit4D double mutant."This is confusing wording because there is not an increase in Oxygen Consumption relative to WT in the mct1del sit4del double mutant, but rather a decrease. I assume the authors mean increase with respect to the mct1del mutant.

We rephrased this sentence. It now reads “… is insufficient to explain the profound increase in MMP observed in the *mct1*D *sit4*D double mutant in comparison to *mct1*D single mutant.”

In Figure S2F it would be helpful to label the F0 component of the ATP synthase if that is what disappears in the rpo41del strains. I was a bit confused because the antibody for Atp2p, part of the F1 component was shown.

We do not have the antibody for any component in Fo subunit, but the molecular weight on our BN-PAGEs corresponds to the expected full ATP synthase complex. We thus used Atp2 to indicate ATP synthase abundance.

Figure 2F is a bit hard to interpret because the authors suggest that the mitochondrial membrane potential is not increased by reversal of the ATP synthase. After reading through the relevant text it seems like what the authors want to say is that they have ruled out the reversal of ATP synthase as a driver of the increased MMP in sit4del yeast. The dark arrow towards Mitochondrial Membrane potential makes it look like reversal of the ATP synthase is contributing to the increase of mitochondrial membrane potential. Perhaps it would be easier to interpret if there was a red cross below reversal of the ATP Synthase as well?

We appreciate this suggestion and now use dotted arrow to indicate that this is a theoretical way of generating MMP but is not utilized in *sit4*D cells.

Figure 4B shows fold increases with and without Phosphate – that would also be useful to show on other such comparisons (1A, 1B, 2C, 2D).

We now display all relevant fold changes in Figure 1A, 1B, 2C, and 2D.

Figure 4C. Caption does not list both drugs.

We indeed only used bongkrekic acid for this experiment. There is no complex III in *rho^0^* cells. Therefore, treating *rho^0^* cells with bongkrekic acid alone was enough to abolish the MMP increase.

Figure 4D: It would be helpful to show BKA on this schematic.

We included BKA on the schematic as suggested.

While deletion of PHO85 in combination with deletion of MCT1 helps to establish a link between phosphate starvation and mitochondrial integrity (FiguresS4B-D) it would be more direct to test phosphate starvation directly in an mct1del background.

We think including more data such as phosphate starvation in *mct1*D cells will be repetitive and will not add much additional information to the current story. We want to do our best to keep the story concise, yet thorough, and only show essential data to support our model.

The phrase "The scope of responses elicited in cells experiencing high MMP, however, has not been previously interrogated." is hard to evaluate. It seems from the authors previous sentence that a lot has already been done to interrogate the scope of responses to high MMP. It is also not obvious that this study fully interrogates the scope of responses. This statement may need to be qualified or made more clear.

We should have been more specific with the previous sentence and relevant citations. All of those literatures describe responses to reduced MMP. We rephased these sentences to emphasize that compared to the cellular responses to low MMP, we lack understanding of how cells respond to high MMP. This sentence now reads “The scope of responses elicited in cells experiencing high MMP, however, has not been previously interrogated nearly to the same extent as cells with reduced MMP.”

In the methods section for the Mitochondrial Protein Import Assay it is unclear what '10OD of total culture at an OD600 between 0.3 and 0.5' means. Approximate total number of cells or a volume should be provided.

We added an estimate of cell number in the method section. 10 OD corresponds to roughly 10^8^ cells in this case.

Reference

Arndt, K. T., Styles, C. A., and Fink, G. R. (1989). A suppressor of a HIS4 transcriptional defect encodes a protein with homology to the catalytic subunit of protein phosphatases. *Cell*, *56*(4), 527–537. https://doi.org/10.1016/00928674(89)90576-X

Dimmer, K. S., Fritz, S., Fuchs, F., Messerschmitt, M., Weinbach, N., Neupert, W., and Westermann, B. (2002). Genetic basis of mitochondrial function and morphology in *Saccharomyces cerevisiae*. *Molecular Biology of the Cell*, *13*(3), 847–853. https://doi.org/10.1091/mbc.01-12-0588

Gupta, R., Walvekar, A. S., Liang, S., Rashida, Z., Shah, P., and Laxman, S. (2019). A tRNA modification balances carbon and nitrogen metabolism by regulating phosphate homeostasis. *ELife*, *8*, e44795. https://doi.org/10.7554/*eLife*.44795

Jablonka, W., Guzmán, S., Ramírez, J., and Montero-Lomelí, M. (2006). Deviation of carbohydrate metabolism by the SIT4 phosphatase in *Saccharomyces cerevisiae*. *Biochimica et Biophysica Acta (BBA) – General Subjects*, *1760*(8), 1281–1291. https://doi.org/10.1016/j.bbagen.2006.02.014

Liu, N.-N., Flanagan, P. R., Zeng, J., Jani, N. M., Cardenas, M. E., Moran, G. P., and Köhler, J. R. (2017). Phosphate is the third nutrient monitored by TOR in *Candida albicans* and provides a target for fungal-specific indirect TOR inhibition. *Proceedings of the National Academy of Sciences*, *114*(24), 6346–6351. https://doi.org/10.1073/pnas.1617799114

Sutton, A., Immanuel, D., and Arndt, K. T. (1991). The SIT4 protein phosphatase functions in late G1 for progression into S phase. *Molecular and Cellular Biology*, *11*(4), 2133–2148.